# FPDou: Mastering DouDizhu with Fictitious Play

## Abstract

DouDizhu is a challenging three-player imperfect-information game involving competition and cooperation. Despite strong performance, existing methods are primarily developed with reinforcement learning (RL) without closely examining the stationary assumption. Specifically, DouDizhu's three-player nature entails algorithms to approximate Nash equilibria, but existing methods typically update/learn all players' strategies simultaneously. This creates a non-stationary environment that impedes RL-based best-response learning and hinders convergence to Nash equilibria. Inspired by Generalized Weakened Fictitious Play (GWFP), we propose FPDou. More specifically, to ease the use of GWFP, we adopt a perfect-training-imperfect-execution paradigm: we treat the two Peasants as one player by sharing information during training, which converts DouDizhu into a two-player zero-sum game amenable to GWFP's analysis. To mitigate the training-execution gap, we introduce a regularization term to penalize the policy discrepancy between perfect and imperfect information. To make learning efficient, we design a practical implementation that consolidates RL and supervised learning into a single step, eliminating the need to train two separate networks. To address non-stationarity, we alternate on-policy/off-policy updates. This not only preserves stationarity for $\epsilon$-best-response learning but also enhances sample efficiency by using data for both sides. FPDou achieves a new state of the art: it uses a $3\times$ smaller model without handcrafted features, outperforms DouZero and PerfectDou in both win rate and score, and ranks first among 452 bots on the Botzone platform. The anonymous demo and code are provided for reproducibility.

## 1 Introduction

DouDizhu (Fighting the Landlord) is a three-player imperfect-information game that involves both competition and cooperation. Its strategic complexity and large state-action space have made it a popular benchmark in artificial intelligence research (Zhang et al., 2024). Early approaches based on rule-based systems (Zha et al., 2021a) and supervised learning (Li et al., 2019; Tan et al., 2021) achieved limited performance. Over the years, advancements in self-play methods combined with deep RL (Sutton & Barto, 2018) have been introduced to mastering the game, yielding significant performance improvements (Jiang et al., 2019; Zha et al., 2021b; Yang et al., 2022). Despite these successes achieved by integrating various RL techniques, we argue that one key overlooked aspect is that: in *multi-player games*, simultaneous adaptation of all players' strategies based on recently played games violates the *stationary* assumption in standard RL. In fact, in games, it is known that *best-response* learning is often essential for algorithm convergence (Bowling, 2004; Shamma & Arslan, 2005; Xu, 2016; Gao et al., 2018). Yet in DouDizhu, this theoretical result has been largely neglected due to the empirical success of directly introducing RL techniques to the game.

Motivated to fill this gap, we introduce FPDou in this paper: a self-play method developed by more strictly following game-theoretic frameworks of fictitious play (FP) (Brown, 1951; Robinson, 1951) and generalized weakened fictitious play (GWFP) (Leslie & Collins, 2006). To ease learning for DouDizhu and align it with GWFP analysis, we adopt a *perfect-training-imperfect-execution* with regularization. This reduces DouDizhu to a *two-player zero-sum* game in training: the two Peasants act like one player by sharing card information, competing against the Landlord with efficient coordination from the beginning. To properly adapt GWFP, we analyze GWFP's process and propose to use a simplified form. Based on this form, we design a practical implementation of FPDou that consolidates RL and supervised learning into a single step—eliminating the need to train two separate

networks. This design not only reduces computational complexity but also facilitates the practical deployment of GWFP for DouDizhu.

We next describe the detailed design for FPDou. First, by treating the two Peasants as a unified player, FPDou updates their policies together. To mitigate the gap between training and testing, a regularization term is used, ensuring that the agent can approximate the perfect-training policy at test time when the model only has access to imperfect observations. Second, FPDou alternates between updating Peasants-team and Landlord across iterations. One side is trained on-policy in a stationary environment, while a copy of the other side is trained off-policy using data from the replay buffer. This on-policy/off-policy alternation ensures each side learns an $\epsilon$-best response in a stationary environment—fulfilling GWFP's requirement for best-response learning within an RL framework—while improving data efficiency by leveraging data to update both sides concurrently. Finally, we employ a distributional Q-network to model value distributions, enabling flexible evaluation across different objectives, such as Winning Percentage (WP) and Average Difference in Points (ADP) (Jiang et al., 2019), without retraining.

Due to a principled algorithm design that more closely follows the spirit of GWFP, FPDou achieves state-of-the-art performance with remarkable efficiency: despite a $3\times$ smaller model, it outperforms all competitors on both WP and ADP when trained on a single server (32 CPUs, 6 GPUs). It surpasses RLCard (Zha et al., 2021a) in 30 minutes, SL (Zha et al., 2021b) in 5 hours, and the strongest baselines (DouZero, DouZero-WP (Zha et al., 2021b), PerfectDou (Yang et al., 2022)) within 5–20 days. Our experiments reveal several interesting findings: (1) The strength shifts during training. The Landlord exhibits dominance over the Peasants during early training. As training progresses, the Peasants learn to cooperate and surpass the Landlord. (2) Explicit exploration appears unnecessary. This may be attributed to the high diversity in game initialization–more pronounced than in Atari and Go–and the effect of policy churn (Schaul et al., 2022), which provides implicit exploration. (3) The first action is pivotal. A poor opening move significantly lowers the chance of winning, even if the player makes their best possible decisions thereafter.

## 2 BACKGROUND

### 2.1 GAME OF DOUDIZHU

DouDizhu is a shedding-type game, where players aim to be the first to empty their hand cards. The game features asymmetric roles: one player is the Landlord, while the other two form a cooperative team as the Peasants. The players cannot communicate. The game consists of two phases: bidding and card-playing. In the bidding phase, players bid to be the Landlord, who then receives three additional face-down cards. The others form a team as Peasants. In the card-playing phase, players take turns counterclockwise, starting with the Landlord. In each round, the first player can play any valid card combination (e.g., single, pair). Players must either pass or play a stronger combination of the same type, or a bomb/rocket that beats all other combinations. Each bomb or rocket doubles the final score. A round ends after two consecutive passes, and the player who last played leads the next round. After the game, the winner side (Peasants or Landlord) receives a reward from the other side. As in previous work, we skip the bidding phase and focus on training for the card-playing phase. More detailed description on game rules are provided in Section B.

### 2.2 PRELIMINARIES ON GAME THEORY

**Markov Decision Process (MDP), Markov Games and Reinforcement Learning.** For single-agent sequential decision-making, a *Markov Decision Process (MDP)* (Bellman, 1957) is defined by a tuple $\langle \mathcal{S}, \mathcal{A}, P, R, \gamma \rangle$. $\mathcal{S}$ is the state space, $\mathcal{A}$ is the action space, $P(s'|s, a)$ is the transition probability, $R(s, a)$ is the reward function, $\gamma \in [0, 1]$ is the discount factor. The goal is to learn a policy that maximizes the expected cumulative rewards $\mathbb{E}[\sum_{t=0}^{\infty} \gamma^t r_t]$. For multi-agent interactions (e.g., DouDizhu), *Markov Games* (Littman, 1994) (or *Stochastic Games*) generalize MDPs to $n$ players, defined by $\langle \mathcal{N}, \mathcal{S}, \mathcal{A}, P, R, \gamma \rangle$. $\mathcal{N} = \{1, \cdots, n\}$ is the set of players. $\mathcal{A} = \times_{i \in \mathcal{N}} \mathcal{A}^i$ is the joint action set, with $\mathcal{A}^i$ being player $i$'s action set. $P(s'|s, a)$ is the transition probability, dependent on the joint action profile $a = (a^1, \cdots, a^n)$. $R(s, a) = (R^1(s, a), \cdots, R^n(s, a))$ is the reward vector, with $R^i(s, a)$ being player $i$'s reward. Markov games model dynamic interactions where players' actions jointly shape state transitions and rewards. Reinforcement learning (Sutton & Barto, 2018) refers to a learning process where an agent learns the optimal policy by interacting with an environment. Most RL algorithms are developed with MDP formulation due to *stationary* property of MDPs. When applying RL to games, the terms *policy* and *strategy* are often used interchangeably.

**Normal-Form and Extensive-Form Games** (Fudenberg & Tirole, 1991; Osborne & Rubinstein, 1994). Markov games can be connected to two classic game categories, static (normal-form) and sequential (extensive-form), based on how their interactive processes are structured. A *normal-form* game, which ignores $\mathcal{S}$ and $P$, is defined by $\langle \mathcal{N}, \mathcal{A}, R \rangle$. It is played in a single round, with players selecting actions simultaneously. The game is *zero-sum* if the rewards sum across all players is zero, i.e., $\sum_{i \in \mathcal{N}} R^i(a) = 0 \,\forall\, a \in \mathcal{A}$. An *extensive-form* game involves multiple rounds. It aligns with the tuple structure of Markov games, though it does not emphasize the Markov property. Players take turns making decisions, and the game progresses through a sequence of states. The game is *imperfect-information* if players lack full access to the game state. In this case, each player must act based on an *information set* $u \in \mathcal{U}$, which contains states that a player cannot distinguish between.

**Strategy**. A strategy $\pi^i \in \Delta(\mathcal{A}^i)$ is a complete plan for player $i$, specifying an action for every possible state. Here, $\Delta(\mathcal{A}^i)$ denotes the set of probability distributions over the action set $\mathcal{A}^i$. A *pure strategy* deterministically selects one action per state, while a *mixed strategy* is a distribution over pure strategies. In extensive-form games, players typically use *behavior strategies* $\pi^i(u) \in \Delta(\mathcal{A}^i(u))$, which assign independent action distributions at each information set $u \in \mathcal{U}^i$. Given a *strategy profile* $\pi = (\pi^1, \ldots, \pi^n)$, a *best response* $b^i(\pi^{-i})$ is a strategy for player $i$ that maximizes its expected reward against $\pi^{-i}$,

$$b^i(\pi^{-i}) = \arg\max_{\pi^i} R^i(\pi^i, \pi^{-i}). \tag{1}$$

Here, $\pi^{-i}$ denotes the strategy profile of all players except $i$. We slightly overload $R$: it denotes immediate reward when applied to states or action profiles, and expected reward for strategies.

**Nash Equilibrium**. A Nash equilibrium (Nash, 1950; 1951) is a strategy profile $\pi^* = (\pi^{*,1}, \ldots, \pi^{*,n})$ where no player $i$ can unilaterally improve its expected reward by deviating from the strategy $\pi^{*,i}$, given the strategies of others $\pi^{*,-i}$. Formally, $\pi^*$ is a Nash equilibrium if for each player $i \in \mathcal{N}$, $\pi^{*,i}$ is a best response to $\pi^{*,-i}$, i.e.,

$$R^i(\pi^*) \geq R^i(\pi^i, \pi^{*,-i}), \quad \forall \pi^i \in \Delta(\mathcal{A}^i). \tag{2}$$

Thus, learning best responses for all players is a common approach to finding Nash equilibria.

# 3 APPLYING FICTITIOUS PLAY TO DOUDIZHU

Fictitious play (FP) (Brown, 1951; Robinson, 1951; Berger, 2007) is a self-play algorithm in which each player iteratively computes best responses to the empirical average of opponents' strategies. We first review an extended variant of FP, generalized weakened fictitious play (GWFP) (Leslie & Collins, 2006), and then examine its applicability to DouDizhu. The key finding is that the primitive form of GWFP is better suited for neural network-based policy learning in large-scale games such as DouDizhu. Based on this insight, we design FPDou, an RL algorithm for DouDizhu that adheres closely to the spirit of GWFP while maintaining high sample efficiency.

## 3.1 GENERALIZED WEAKENED FICTITIOUS PLAY

GWFP extends FP by allowing approximate best responses and tolerating learning errors, while still ensuring convergence to a Nash equilibrium in two-player zero-sum games (Leslie & Collins, 2006).

**Definition 3.1. Generalized Weakened Fictitious Play (GWFP)** (Leslie & Collins, 2006) is a process $\{\pi_t\}$, $\pi_t \in \times_{i \in \mathcal{N}} \Delta(\mathcal{A}^i)$, such that for each player $i \in \mathcal{N}$ and time step $t \geq 0$, the strategy update is given by:

$$\pi_{t+1}^i \in (1 - \alpha_{t+1})\pi_t^i + \alpha_{t+1}(b_{\epsilon_t}^i(\pi_t^{-i}) + M_{t+1}^i), \forall i \in \mathcal{N}, t \geq 0, \tag{3}$$

with $\alpha_t \to 0$ and $\epsilon_t \to 0$ as $t \to \infty$, $\sum_{t=1}^{\infty} \alpha_t = \infty$, and $\{M_t\}$ a sequence of perturbations such that $\forall\, T > 0, \lim_{t \to \infty} \sup_k \left\{ \left\| \sum_{i=t}^{k-1} \alpha_{i+1} M_{i+1} \right\| \text{ s.t. } \sum_{i=t}^{k-1} \alpha_{i+1} \leq T \right\} = 0$.

Here, $b_\epsilon^i$ denotes a $\epsilon$-*best response*, which relax best response with a margin $\epsilon$:

$$b_\epsilon^i(\pi^{-i}) \in \{\pi^i : R^i(\pi^i, \pi^{-i}) \geq R^i(b^i(\pi^{-i}), \pi^{-i}) - \epsilon\}. \tag{4}$$

FP is recovered by setting $\epsilon_t = M_t = 0$ and $\alpha_t = 1/t$. The use of $\epsilon$-best responses makes GWFP practical in large-scale domains where exact best responses are intractable, and the perturbation term $M_t$ accommodates estimation errors arising from function approximation (Heinrich et al., 2015).

In Definition 3.1, $\pi_t$ is defined over mixed strategies, which can be implemented as weighted combinations of realization-equivalent behavioral strategies under perfect recall (Heinrich et al.,

2015). Two strategies are *realization-equivalent* if they induce the same distribution over information sets (Kuhn, 1953). *Perfect recall* means each player remembers the history $\{u_1^i, a_1^i, \cdots, u_k^i\}$ that led to the current information set $u_k^i$. Behavioral strategies allow players to make decisions based on local policies at each information set, which naturally aligns with how deep RL agents are trained. The behavioral strategy formulation preserves GWFP's convergence guarantees and enables scalable implementations using neural networks. These insights establish GWFP as a theoretically well-founded yet practical self-play framework for complex domains (Heinrich & Silver, 2016; Vinyals et al., 2019; Berner et al., 2019; Zha et al., 2021b; Yang et al., 2022).

## 3.2 EQUIVALENT FORM OF GWFP FOR BETTER APPLICABILITY

Directly implementing GWFP in deep RL is inefficient. It typically requires maintaining and training two separate networks per player: one for the average policy $\pi_t^i$ (via supervised learning) and another for the new best response $b_{\epsilon_t}^i(\pi_t^{-i})$ (via RL) (Heinrich et al., 2015; Heinrich & Silver, 2016; Cloud et al., 2023). This approach of learning two distinct networks is computationally demanding.

To create a more practical algorithm, the recursive update rule of GWFP can be expanded into a non-recursive average over the sequence of $\epsilon$-best responses. For the common choice of $\alpha_t = 1/t$, the next average policy $\pi_{t+1}^i$ has the following form:

$$\pi_{t+1}^i = \underbrace{\frac{1}{t+1}\pi_0^i + \sum_{k=1}^{t}\left(\frac{1}{t+1}b_{\epsilon_{k-1}}^i(\pi_{k-1}^{-i})\right)}_{\text{Past } \epsilon\text{-Best Responses}} + \underbrace{\frac{1}{t+1}b_{\epsilon_t}^i(\pi_t^{-i})}_{\text{New } \epsilon\text{-Best Response}} . \tag{5}$$

The perturbation term $M_t$ is omitted here for clarity, and a full derivation is provided in Section D.2.

As $t$ increases, the influence of the initial strategy $\pi_0^i$ fades and has no impact on convergence. Moreover, since no prior strategy exists before $\pi_0^i$, $\pi_0^i$ can be viewed as an $\hat{\epsilon}$-best response to an unknown opponent strategy (with a sufficiently large $\hat{\epsilon}$)—consistent with the semantics of the $\epsilon$-best response sequence. This means the right-hand side of Eq. (5) consists solely of an average over a sequence of $\epsilon$-best responses, all sharing consistent input-output semantics. This consistency is the key insight underpinning our algorithm's design. It allows us to consolidate the two-step process into one step: each player uses one network $\pi_t^i$, eliminating the need to maintain two separate networks for $\pi_t^i$ and $b_{\epsilon_t}^i(\pi_t^{-i})$. Specifically, our FPDou agent uses one network to learn the **new $\epsilon$-best response** via on-policy RL against the current fixed opponent, and simultaneously learns the average of the **past $\epsilon$-best responses** by training on historical data sampled from a replay buffer. This formulation provides a practical and efficient implementation of GWFP for large-scale games.

## 3.3 FPDOU: PRACTICAL AGENT FOR DOUDIZHU WITH FICTITIOUS PLAY

Returning to DouDizhu, we find that one commonality of existing methods such as DouZero (Zha et al., 2021b) and PerfectDou (Yang et al., 2022) is that they use deep RL to learn best responses but update all players' policies simultaneously. This scheme, although it works in practice with some success, breaks the key stationary assumption of RL since the opponent's policy is changing at the same time. Simultaneous strategy update is also inconsistent with the spirit of GWFP: each player's policy may not be an $\epsilon$-best response to others, causing a convergence issue (Bowling, 2004; Gao et al., 2018). Furthermore, learning only against the latest strategy without averaging the historical sequence failed to obey the history averaging principle of FP. Motivated by applying the core ideas of GWFP for a better agent, we design FPDou—a method that is aspired to be not only more principled in theory but also remains highly sample efficient in practice. The key components are as follows:

**Perfect Training with Regularization and Imperfect Execution**. The two peasants are treated as a unified player against the Landlord during training. To bridge the gap between training and testing, we introduce a regularization term during training. The objective is to encourage consistency between decisions made with and without access to perfect information.

**Learning $\epsilon$-best response and average strategy**. To ensure $\epsilon$-best response learning and to improve sample efficiency, we adopt an alternating training scheme. When training the Peasants, the Landlord's policy is fixed, and the Peasants are trained on-policy to learn an $\epsilon$-best response. Meanwhile, a copy of the Landlord is updated off-policy using data from the replay buffer. Once the Peasants' policy is learned, we switch roles: the Landlord is trained on-policy against fixed Peasants, while a copy of the Peasants is updated off-policy. On-policy learning against fixed opponents ensures

a stationary environment; it also enables direct assessment of whether an $\epsilon$-best response has been achieved, eliminating the need for separate policy evaluation. Off-policy learning of the other side improves sample efficiency by enabling data use for both sides. This alternating scheme balances theoretical soundness with practical sample efficiency.

To approximate the average $\epsilon$-best response sequence $\{b^i_{\epsilon_{k-1}}(\pi^{-i}_{k-1})\}$ in Eq. (5)–without maintaining parameters of historical strategies–we store trajectories collected from past best responses in a replay buffer and learn the average over this mixture. Thus each time we not only learn from recently collected trajectories, but also learn from historical data. This averaging is performed alongside $\epsilon$-best response learning simply by sampling from the buffer, analogous to AlphaGo that maintains a buffer and conducts supervised learning (Silver et al., 2016; 2017). This unified process enables us to directly approximate the next average policy $\pi^i_{t+1}$: simultaneously learning the new $\epsilon$-best response from newly collected data and the average of past responses from buffer-stored data.

**Distributional Learning**. We use a distributional Q-network (Bellemare et al., 2017) to model the value distribution. It enables us to compute different objectives at test-time, such as Winning Percentage (WP) and Average Difference in Points (ADP), without retraining.

## 4 IMPLEMENTATION OF FPDOU SYSTEM

We present the implementation details of FPDou. Specifically, Section 4.1 describes the card representation and network architecture, designed to satisfy perfect recall and support multiple evaluation objectives. Section 4.2 contains the training details, including $\epsilon$-best response learning, information sharing between Peasants, and the distributed training pipeline.

### 4.1 CARD REPRESENTATION AND NETWORK ARCHITECTURE

Both states and actions are represented using one-hot $4 \times 15$ matrices, which encode the number of cards for each rank. Fig. 1 illustrates the hand representation. Since each deck has only one big joker and one small joker, we mark the six unused entries in the last two columns as 1 to represent the "Pass" action. To approximate perfect recall, each state stacks the current hand with the previous 60 actions, as 99% of games are completed within 60 steps (Section G Table 7). Unlike prior works (Jiang et al., 2019; Yang et al., 2022), we avoid human-designed features and rely solely on raw card and action history. Details of each channel are provided in Section E.2 Table 3.

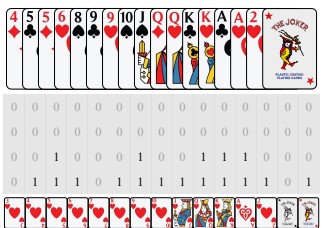

Figure 1: Hand representation.

Consistent with prior works (Zha et al., 2021b; Yang et al., 2022), three networks are maintained (one for each position). Each network is a distributional Q-network (Bellemare et al., 2017), implemented as a CNN with skip connections (He et al., 2016), and takes state-action pairs as input. Compared to fully connected networks, CNNs al-

Table 1: FPDou has the smallest model.

| Model | DouZero / SL | PerfectDou | FPDou |
|---|---|---|---|
| Size | 18 MB | 13.4 MB | 4.5 MB |

low efficient extension of action history without significantly increasing model size. FPDou is only 4.5 MB, significantly smaller than DouZero and PerfectDou (Table 1). The output is a distribution over Q-values, represented using 8 bins that correspond to win/loss with different numbers of bombs (0, 1, 2, or $\geq 3$), mapped to rewards of $r = [-4, -3, -2, -1, 1, 2, 3, 4]$ [1]. Since outcomes with $\geq 3$ bombs occur in fewer than 0.5% of games (Section G Fig. 8), 8 bins suffice to approximate the return distribution. Detailed network architecture is provided in Section E.3.

### 4.2 TRAINING DETAILS

We begin with the Q-network training and feature regularization, then describe their integration into the GWFP framework, and finally present the complete distributed implementation.

#### 4.2.1 DEEP REINFORCEMENT LEARNING USING MONTE CARLO ESTIMATION

While Monte Carlo methods are often criticized for high variance, 99% of DouDizhu games finish within 60 steps, which is far shorter than games such as Go (Silver et al., 2016) and Atari (Bellemare et al., 2013) with hundreds or thousands of steps. As a result, Monte Carlo estimation in DouDizhu (Zha et al., 2021b) suffers less from high variance and introduces no estimation bias.

---

[1]Since DouDizhu provides only a final reward, we use the terms *reward* and *return* interchangeably.

Figure 2: FPDou training pipeline. (**Left**) Multiple actor workers collect data into replay buffers. (**Right**) A centralized learner samples from all buffers to train global Q-networks. Only the side currently learning its $\epsilon$-best response synchronizes parameters from the learner.

To collect data, we adopt a near-greedy exploration strategy, `top-k@n`, which samples the top $k = 3$ actions with the highest Q-values for the first $n = 3$ steps (i.e., one step per player) and then switches to greedy. We wait until the end of the game, assign the final reward to all transitions along the trajectory, and store them in the replay buffer. During Landlord training, we sample batches of transitions $\{(s_i, a_i, y_i)\}_{i=1}^{B}$, where $y_i$ is a one-hot vector of the game reward. Let $(p, z) = f_\theta(s, a)$ denote the network output: $p$ is the predicted Q-distribution, and $z$ is the output of the penultimate layer. The training objective minimizes cross-entropy loss:

$$\mathcal{L} = -\frac{1}{B} \sum_{i=1}^{B} y_i^\top \log p_i, \quad \text{where } (p_i, \_) = f_\theta(s_i, a_i). \tag{6}$$

During Peasants training, we additionally provide perfect observation to Peasants to promote cooperation, enabling them to act as a unified player. To recover this policy when only using imperfect observation at test time, we introduce a regularization term. Specifically, with $s$ denoting a Peasant's imperfect observation and $\bar{s}$ the augmented perfect observation including the other Peasant's hand, we regularize the output features $z$ and $\bar{z}$ uses an L2 loss:

$$\mathcal{L} = \frac{1}{B} \sum_{i=1}^{B} (-y_i^\top \log p_i + \|z_i - \bar{z}_i\|^2), \quad \text{where } (p_i, z_i) = f_\theta(s_i, a_i), (\_, \bar{z}_i) = f_\theta(\bar{s}_i, a_i). \tag{7}$$

Here, we include both $s_i$ and $\bar{s}_i$ in a single batch, so training time is not doubled. Note that $\bar{s}_i$ is only used during training; during test time, we only provide the imperfect $s_i$. Notably, the Q-network enables the integration of reinforcement learning and supervised learning. When interacting with the environment and learning from newly collected data, it corresponds to the policy iteration of reinforcement learning; when learning from buffer-stored historical data, it learns the average of past $\epsilon$-best responses. Thus, the Q-network allows us to unify these two processes under the same loss function, with the only distinction lying in the data source.

### 4.2.2 $\epsilon$-BEST RESPONSE IN FICTITIOUS PLAY

To ensure learning an $\epsilon$-best response at each iteration, we track results from the most recent 200 games and apply a win-rate threshold $\tau \in (0, 1)$ to determine whether the current policy qualifies as an $\epsilon$-best response. Once the threshold is met, the on-policy training switches to the other player. Since the game's Nash value is unknown, we set $\tau = 0.5$: assuming a balanced game, and aligning with fuzzy measurement practice, where 0.5 (distinct from values near 0 or 1) represents the membership degree for high uncertainty (Verma & Kumar, 2020; Wan & Yi, 2015). Early in training, when the opponent is weak, achieving a win rate meeting $\tau$ implies an $\epsilon_t$-best response with $\epsilon_t > 0$. As training progresses and all players improve, approaching $\tau$ implies a decreasing $\epsilon_t$; ideally, $\epsilon_t \to 0$ under the balanced game assumption and align with GWFP's condition in Definition 3.1. This learning scheme is incorporated into the GWFP process in Eq. (5). Besides this fixed threshold, we also conducted a complementary experiment with adaptive threshold adjustment during training; details are in Section G.5. Moreover, Section F further discusses approximations in our algorithm design when adapting GWFP to a practical method for DouDizhu.

### 4.2.3 DISTRIBUTED TRAINING PIPELINE

To accelerate training, we implement a distributed system using multiple GPUs. The system consists of parallel actor workers and a centralized learner. Each actor maintains three local Q-networks

(one for each position) to interact with the environment and collect trajectories into a replay buffer. The learner maintains global Q-networks for all three positions and samples data from all replay buffers to update the models. At each iteration, only the player currently learning its $\epsilon$-best response synchronizes with the learner to fetch the latest parameters; the other players use fixed parameters from the previous iteration to ensure a stationary environment. Experiments were conducted on a machine equipped with 2 AMD EPYC 7313 CPUs (16 cores, 32 threads in total) and 6 NVIDIA GPUs (1 RTX A5000, 2 TITAN RTX, and 3 RTX 2080Ti). One GPU is dedicated to the learner, while the remaining five GPUs serve actor workers, with two actors per GPU. Given the heterogeneous GPU setup, we aggregate training batches across all replay buffers to mitigate performance discrepancies caused by varying GPU speeds. Training lasted for one month. Fig. 2 shows the training pipeline, with pseudocode and hyperparameters provided in Section E.

## 5 EXPERIMENTS

**Evaluation Metrics.** We evaluate FPDou using the RLCard environment (Zha et al., 2021a), following the same protocol and metrics as in the previous work (Zha et al., 2021b; Yang et al., 2022). Specifically, we adopt two widely used metrics:(1) Winning Percentage (**WP**), the proportion of games won. WP > 0.5 indicates better performance. (2) Average Difference in Points (**ADP**), the average per-game point difference between two methods. The base point is 1. Each bomb doubles the point. ADP > 0 indicates better performance. For evaluation, we randomly generate 10,000 decks and have each pair of methods compete on them. For fair comparison, each deck is played twice: model A first plays as Landlord and B as Peasants, then they switch roles and replay the same deck.

**Baseline Methods.** We compare FPDou with recent and state-of-the-art (SoTA) open-source models: **PerfectDou** (Yang et al., 2022)—uses actor-critic with "perfect training, imperfect execution" and a heuristic reward function; **DouZero** (Zha et al., 2021b)—uses Deep Monte Carlo (DMC) optimized for ADP; **DouZero-WP**—DouZero trained for WP; **SL** (Zha et al., 2021b)—supervised agent trained on 49 million expert games; **RLCard** (Zha et al., 2021a)—simple rule-based agent. For reference, we also include the comparison to **Random**—an agent that uniformly samples actions.

Table 2: Performance of FPDou against baselines over 10,000 decks, rounded to three decimal places. A outperforms B if WP> 0.5 or ADP> 0 (in boldface). Methods are ranked based on ADP.

| Rank | B / A | FPDou | | PerfectDou | | DouZero | | DouZero-WP | | SL | | RLCard | | Random | |
|---|---|---|---|---|---|---|---|---|---|---|---|---|---|---|---|
| | | WP | ADP | WP | ADP | WP | ADP | WP | ADP | WP | ADP | WP | ADP | WP | ADP |
| 1 | FPDou | - | - | **0.520** | **0.100** | **0.562** | **0.197** | **0.510** | **0.333** | **0.684** | **0.996** | **0.894** | **2.522** | **0.993** | **3.107** |
| 2 | PerfectDou | 0.480 | -0.100 | - | - | **0.543** | **0.141** | 0.489 | **0.212** | **0.669** | **1.033** | **0.890** | **2.495** | **0.993** | **3.087** |
| 3 | DouZero | 0.439 | -0.197 | 0.457 | -0.141 | - | - | 0.453 | **0.119** | **0.611** | **0.774** | **0.857** | **2.377** | **0.987** | **3.043** |
| 4 | DouZero-WP | 0.490 | -0.333 | **0.511** | -0.212 | **0.548** | -0.119 | - | - | **0.660** | **0.715** | **0.884** | **2.164** | **0.988** | **2.741** |
| 5 | SL | 0.316 | -0.996 | 0.331 | -1.033 | 0.389 | -0.774 | 0.340 | -0.715 | - | - | **0.808** | **1.787** | **0.974** | **2.696** |
| 6 | RLCard | 0.106 | -2.522 | 0.110 | -2.495 | 0.144 | -2.377 | 0.116 | -2.164 | 0.192 | -1.787 | - | - | **0.942** | **2.504** |
| 7 | Random | 0.007 | -3.107 | 0.007 | -3.087 | 0.013 | -3.043 | 0.012 | -2.741 | 0.026 | -2.696 | 0.058 | -2.504 | - | - |

### 5.1 SUPERIOR PERFORMANCE OVER BASELINES

As shown in Table 2, FPDou ranks first in both WP and ADP. It achieves a 52% WP and 0.1 ADP against PerfectDou, with larger performance improvements over other baselines. Regarding training time to surpass baselines: For WP, FPDou surpasses RLCard within 30 minutes, SL in 5 hours, DouZero in 2 days, and both PerfectDou and DouZero-WP in 18 days. For ADP, it surpasses RLCard in 30 minutes, SL in 5 hours, DouZero-WP in 5 days, DouZero in 9 days, and PerfectDou in 20 days. Furthermore, FPDou ranks first among 452 bots on the Botzone platform [2], demonstrating its strong performance. Detailed results and learning curves for each position are provided in Section G.

### 5.2 ANALYSIS OF FPDOU

To answer the following four questions, we analyze FPDou's training process over the first $10^9$ steps. **Q1:** How do the CNN design, off-policy learning, and regularization each contribute to the superior performance of FPDou? **Q2:** How does on-policy $\epsilon$-best response learning evolve over training? **Q3:** How effective is the top-k@n exploration strategy in FPDou? **Q4**: How scalable is FPDou w.r.t. different batch sizes and neural network sizes?

---

[2] https://botzone.org.cn/

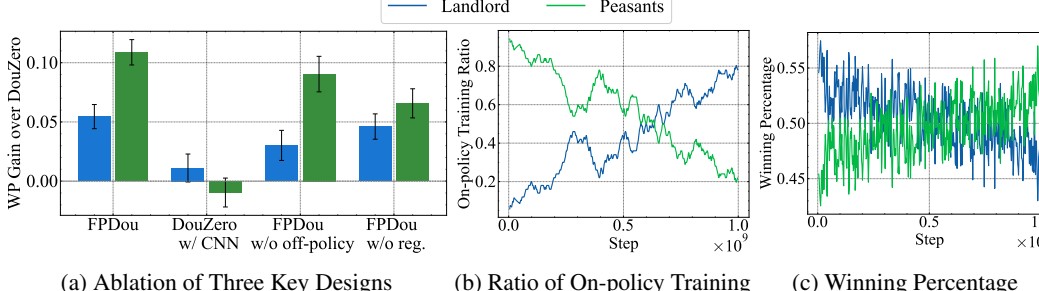

(a) Ablation of Three Key Designs    (b) Ratio of On-policy Training    (c) Winning Percentage

Figure 3: **(a)** Full FPDou achieves the strongest performance. `DouZero w/ CNN` (FPDou's architecture, DouZero's algorithm) achieves similar performance as DouZero but reduces model size from 18 MB to 4.5 MB. `FPDou w/o off-policy` and `FPDou w/o reg.` show reduced performance in comparison to FPDou. **(b)** As training progresses, ratio of on-policy learning computation for Peasants decreases while increases for Landlord. **(c)** As training progresses, Peasants' winning percentage increases while Landlord's decreases.

**How do the CNN design, off-policy learning, and regularization each contribute to the superior performance of FPDou?**    We evaluate four algorithms:(1) full FPDou, (2) `DouZero w/ CNN`—which adopts FPDou's neural network architecture but retains DouZero's algorithmic design, (3) `FPDou w/o off-policy`—which only updates players via on-policy learning, (4) `FPDou w/o reg.`—which removes the regularization term from FPDou. Fig. 3a presents the winning percentages against DouZero for each algorithm both as Peastants and Landlord. Clearly, FPDou achieves the strongest performance, indicating the collective effectiveness of the *off-policy* learning and regularization term. `DouZero w/ CNN` exhibits similar playing strength as DouZero, despite that the model size is reduced from 18 MB to 4.5 MB—this implies that the CNN architecture we designed for FPDou is not only effective but also highly efficient. `FPDou w/o off-policy` shows reduced performance relative to FPDou, highlighting the importance of off-policy updates for fixed players. Removing the regularization term (`FPDou w/o reg.`) degrades the Peasants' performance due to the increased gap between *training* and *testing*, underscoring the importance of an explicit mechanism for handling the disparity of "perfect-training" and "imperfect-execution".

**How on-policy $\epsilon$-best response learning evolves over training?**    To show the training dynamics of FPDou, we plot the computation ratio allocated for *on-policy* learning for both Peasants and Landlord. As shown in Fig. 3b, early in training, there is a high on-policy ratio for Peasants and a low ratio for Landlord. As training progresses, the phenomenon gradually reversed —more and more on-policy learning is performed by the Landlord side. Fig. 3c shows the reason behind Fig. 3b. Early on, it is hard for the Peasants to beat Landlord because of the extra cards given to Landlord, thus Peasants-team has to leverage more on-policy best response learning to compete with the Landlord. Over time, the Landlord faces increasing difficulty maintaining its advantage. Therefore, the on-policy training gradually shifts toward the Landlord. This asymmetric training pattern shows that FPDou's *on-policy* learning is an effective approach that can automatically adjust the learning budget to the side that requires it most.

**How effective is the top-k@n exploration strategy in FPDou?**    We compare the following strategies. `$\epsilon$-greedy`, used in DouZero (Mnih et al., 2015; Zha et al., 2021b), selects a random action with probability $\epsilon$ and the greedy action otherwise. `Greedy` selects the action with the highest Q-value. `Top-k` sampling, widely used in language models (Radford et al., 2019; Liu et al., 2024), samples uniformly from the top $k$ actions. Our method, `top-k@n` sampling, uses top-k sampling for the first $n$ steps before switching to greedy. `softmax@n` relaxes top-k@n by sampling from the full action space upon normalized softmax scores over Q-values for the first $n$ steps, then greedy. We compare these strategies in terms of WP against DouZero with parameters set as follows: $\epsilon = 0.01$ (Zha et al., 2021b), $k = 3$, $n = 3$. Results are shown in Fig. 4a. The greedy policy slightly outperforms $\epsilon$-greedy and clearly outperforms top-k and softmax@n, suggesting that excessive exploration is unnecessary. This could be attributed to the inherent randomness of initial hands, along with policy churn effect in (Schaul et al., 2022), which already introduces sufficient exploration. The softmax@n performs poorly, showing that early-game decisions are critical—one bad move at the start can compromise the entire game. Top-k@n achieves strongest performance.

**How scalable is FPDou w.r.t different batch sizes and neural network sizes?**    It is a question whether we can further enhance FPDou by scaling up the model size and batch size. To see this,

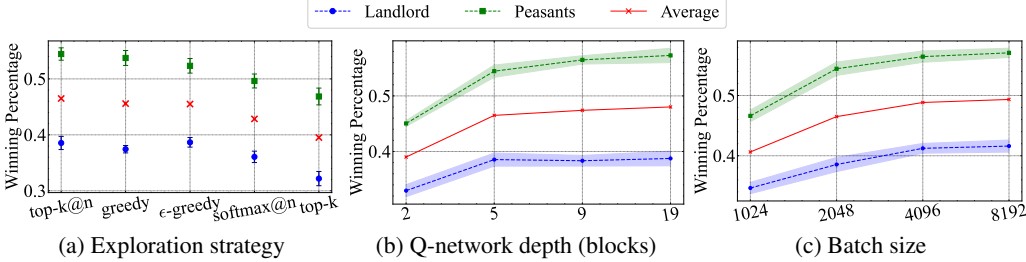

Figure 4: Winning percentage against DouZero under different settings. (a) Greedy policy slightly outperforms $\epsilon$-greedy, while top-k and softmax@n perform poorly, indicating excessive exploration is unnecessary. (b,c) Larger models and batch sizes improve performance.

we try different numbers of residual blocks and batch sizes. As in Figs. 4b and 4c, larger models and batch sizes lead to better performance given the same number of environment steps along with more computation cost and slower training speed on same server. A network with 19 blocks achieves the highest win rate but runs at only 3100 frames per second (FPS)—twice slower than the 5-block version (near 6000 FPS). This result also highlights the scalability of FPDou. Due to resource constraints, our main results are based on a small model (5 blocks) and batch size (2048). Results in Figs. 4b and 4c suggest that further gains might be achieved with larger-scale training for FPDou.

## 6 RELATED WORK

Early approaches for DouDizhu rely on rule-based systems (Zha et al., 2021a) and supervised learning (Li et al., 2019; 2020; Tan et al., 2021); they achieve limited performance and are largely supplanted by deep RL with self-play (Zhang et al., 2024). The first breakthrough is DeltaDou (Jiang et al., 2019), which adopts AlphaZero-style frameworks (Silver et al., 2016; 2017)—combining value networks with Monte Carlo Tree Search (MCTS) (Kocsis & Szepesvári, 2006; Lattimore & Szepesvári, 2020)—and reaches human-level performance for the first time. DeltDou is computationally expensive and sensitive to heuristic quality, a limitation that is common for search-based approaches in games with hidden information (Whitehouse et al., 2011; Zhang et al., 2021). Model-free RL with self-play quickly becomes the dominant paradigm, thanks to its scalability (You et al., 2019; Luo et al., 2022; Yang et al., 2022; Zhao et al., 2022; Wang et al., 2022; Zhao et al., 2023; Yu et al., 2023; Luo & Tan, 2023; Luo et al., 2024; Lei & Lei, 2024). DouZero (Zha et al., 2021b) uses Deep Monte Carlo estimation and Q-networks trained via self-play. PerfectDou (Yang et al., 2022) adopts a sample-efficient actor-critic framework with "perfect training, imperfect execution" and introduces a heuristic reward based on the minimum number of steps to victory. Subsequent enhancements focus on opponent modeling, action pruning, bidding strategies, and training stabilization. These include DouZero+, Full DouZero+, WagerWin, NV-Dou, MDou, RARSMSDou, and OADMCDou, contributing improvements from distributional value decomposition, noise-based exploration, reward shaping to oracle-guided distillation (Zhao et al., 2022; Wang et al., 2022; Yu et al., 2023; Luo et al., 2022; Luo & Tan, 2023; Luo et al., 2024). See Section C for a detailed discussion of these methods.

## 7 CONCLUSION AND LIMITATIONS

We present FPDou, a learning agent that successfully mastered the game of DouDizhu by following the core ideas of GWFP. FPDou approximates DouDizhu via a two-player zero-sum formulation but introduces a novel regularization term for explicitly mitigating the gap between training and testing. It efficiently consolidates reinforcement learning and supervised learning into a single update step and alternates on-/off-policy training between the Peasants team and the Landlord. The experiment results certify that FPDou is not only theoretically plausible but also empirically efficacious, achieving stronger playing strength against state-of-the-art models both in terms of WP and ADP.

While FPDou achieves state-of-the-art performance, several limitations deserve further discussion. Specifically, we adopt a fixed replay buffer for experience storage; considering a large buffer size and the neural networks' memory capacity, $\alpha_t$ is approximately treated as $1/t$, and a more elegant alternative would be to employ reservoir sampling (Vitter, 1985; Heinrich & Silver, 2016). Additionally, our method focuses on training two well-coordinated peasant teammates, as official DouDiZhu competitions are team-based. Whereas in casual community games, zero-shot coordination between unfamiliar teammates is required, which could be an interesting future work.

**Ethics Statement.** This study focuses on the design and evaluation of an AI algorithm (FPDou) for the game of DouDizhu, aiming to propose an effective method for achieving high performance. It involves no human participants, animals, or sensitive data. The game data used for training and testing was synthetically generated by the algorithm itself for research purposes. No ethical risks (e.g., privacy violation, algorithmic bias, or harm to individuals) are associated with this research. All methods were conducted in line with the general ethical principles for academic research in computer science and game AI, and no specific ethical approval was required for this type of study.

**Reproducibility Statement.** An anonymous source code for FPDou is provided at [this link] for reproducibility, as also mentioned in the abstract. Additionally, an anonymous demo for interactive testing is accessible at [this address]. Implementation details and specific hyperparameters are further provided in the appendix, which supports the reliable reproduction of this paper.

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

# Appendix

## Table of Contents

## A    LLM USAGE DISCLOSURE

In this study, Large Language Models (LLMs) were solely used as a general-purpose language refinement tool to enhance the clarity and fluency the manuscript text. The LLM did not participate in any core research processes, including but not limited to research ideation, experimental design, data analysis, result interpretation, or the development of the FPDou algorithm. Its role was limited to polishing the expression of pre-drafted content (e.g., optimizing sentence structure, standardizing academic terminology, and improving logical coherence) without altering the original meaning, data, or conclusions of the research. Thus, the LLM does not qualify as a contributor, and no further substantial role beyond language polishing is disclosed.

## B    THE GAME OF DOUDIZHU

DouDizhu is one of the most popular card games in China, with hundreds of millions of daily active players (Jiang et al., 2019; Zha et al., 2021b). It is a three-player, shedding-type game involving both competition and collaboration under imperfect information. The game is played using a standard 54-card deck, consisting of 15 different ranks: 3, 4, 5, 6, 7, 8, 9, 10 (T), J, Q, K, A, 2, black joker (B), and red joker (R), ranked from lowest to highest. Each of the ranks, except for the jokers, has four cards representing four suits: heart, spade, club, and diamond. The game is composed of two phases: the bidding phase and the card-playing phase.

**Game Setup and Roles.** At the start of each game, 17 cards are dealt to each of the three players, and the remaining 3 cards are placed face-down. These three cards are known as the "Landlord cards" and are awarded to the player who wins the bidding phase. Players take on one of three roles: Landlord, Peasant Up (to the Landlord's left), or Peasant Down (to the Landlord's right). The two Peasants play as a team against the Landlord.

**Bidding Phase.** In the bidding phase, players sequentially bid for the Landlord position based on their private 17-card hands. The three players take turns being the first to bid across multiple games. Bids can be 1, 2, or 3, or a player can pass. The highest bidder becomes the Landlord and receives the three face-down cards, increasing their hand size to 20. The other two players, holding 17 cards each, form a cooperative team. If no player bids, the game is reset. In this paper, we skip the bidding phase and focus on the training of the card-playing phase.

**Card-Playing Phase.** Gameplay proceeds in counterclockwise order starting with the Landlord. The game is played in rounds. In each round, the lead player can play any legal combination of cards, such as a single, pair, triple, chain, bomb (four cards of the same rank), or rocket (a pair of jokers, the highest combination). Subsequent players must either pass or play a combination of the same type with a higher rank. Rockets beat any other combination, and bombs can beat any non-rocket combination, including different hand types. Thus, the rank of the cards is crucial in DouDizhu, while the suit is nearly irrelevant[3].

One round ends when two consecutive players pass. The player who played the last valid hand then starts the next round. The game continues until one player empties their hand, thereby winning the game. If the Landlord plays all their cards first, the Landlord wins and gains a reward from both Peasants. If either Peasant finishes first, both Peasants win and share the reward from the Landlord. The scoring is determined by a base score, multiplied by a dynamic factor increased by each bomb or rocket played. The game is a zero-sum game if we consider the two Peasants as a single player against the Landlord.

**Game Complexity.** DouDizhu is a highly challenging domain for decision-making due to: (1) Imperfect information: players cannot see others' hands. (2) Large state space: approximately $10^{83}$ unique states. (3) Massive and dynamic action space: up to 27,472 legal actions per turn depending on the current hand (Zha et al., 2021a). (4) Asymmetric roles and incentives: the Landlord has more cards and plays first, but faces two coordinated opponents.

Players must reason strategically based on limited observations and the history of plays. The Peasant team must coordinate implicitly without communication to effectively challenge the Landlord, combining elements of cooperation and adversarial play in a partially observable setting.

---

[3]The suit is helpful for inferring whether the Landlord holds the specific bidding card, but a rational player would typically play that suit first if they intend to play a card of the same type.

More thorough details such as card types and scoring rules can be found at pagat.com[4].

## C  EXTENDED RELATED WORK

DouDizhu is a challenging three-player card game characterized by imperfect information and a mix of competition and collaboration. It has long been popular in China and has attracted increasing attention from the AI community in recent decades due to its strategic complexity. Early approaches based on rule-based systems (Zha et al., 2021a) and supervised learning (Li et al., 2019; 2020; Tan et al., 2021) achieved limited performance and have gradually been supplanted by deep reinforcement learning (RL) methods combined with self-play (Zhang et al., 2024) in recent years. These methods have shown significant improvements in performance and scalability, making them more suitable for complex imperfect-information games like DouDizhu.

One research direction follows model-based RL with AlphaZero-style frameworks (Silver et al., 2016; 2017). These methods typically employ Monte Carlo Tree Search (MCTS) (Kocsis & Szepesvári, 2006; Lattimore & Szepesvári, 2020) to explore the game tree and make decisions based on the estimated value of each action. DeltaDou (Jiang et al., 2019) is the first AI agent to achieve top human-level performance in DouDizhu. It proposes Fictitious Play MCTS (FPMCTS), which performs search over imperfect-information trees using Bayesian inference for opponent modeling. To handle the game's large action space, a pre-trained heuristic kicker network is used to prune low-probability actions. While effective, the method's reliance on heuristic abstractions and the computational burden of search and inference make the training process extremely expensive—reportedly taking over two months—and potentially limit performance if the heuristics are suboptimal. These limitations are typical of many search-based methods, which suffer from high computational burden and the difficulty of estimating the hidden information (Whitehouse et al., 2011; Jiang et al., 2019; Zhang et al., 2021).

A more recent and increasingly popular direction is model-free RL with self-play, which improves scalability by replacing explicit search with deep neural networks and sampling-based training (You et al., 2019; Luo et al., 2022; Yang et al., 2022; Zhao et al., 2022; Wang et al., 2022; Zhao et al., 2023; Yu et al., 2023; Luo & Tan, 2023; Luo et al., 2024; Lei & Lei, 2024). DouZero (Zha et al., 2021b) is the most representative work in this category. It adopts Deep Monte Carlo (DMC) estimation, with expressive Q-networks trained purely via self-play, and avoids any domain-specific knowledge. Its simplicity and efficiency enable large-scale training and lay the foundation for numerous follow-up methods. PerfectDou (Yang et al., 2022) builds on DouZero with a sample-efficient actor-critic framework. It adopts a "perfect training, imperfect execution" paradigm: the value network is trained using privileged (perfect) information, while the policy network operates only on partial information during execution. It also introduces a heuristic reward based on the minimum number of steps required to finish a game, serving as an estimate of proximity to victory. While this reward design accelerates early training, it can constrain final performance—a trade-off observed in prior work (Silver et al., 2017; Zha et al., 2021b), and cautioned against by the bitter lesson (Sutton, 2019).

Besides PerfectDou, numerous enhancements to DouZero have been proposed. DouZero+ (Zhao et al., 2022) and Full DouZero+ (Zhao et al., 2023) integrate opponent modeling to infer the actions of other players, and introduce a coach network to select training games with well-matched initial hands while filtering out games that provide limited learning value. Full DouZero+ further extends this framework by modeling the bidding phase, training a dedicated bidding network via Monte Carlo simulation. WagerWin (Wang et al., 2022) decomposes the value function into winning probability, winning Q-value, and losing Q-value, to address the high variance and redundant loss terms in conventional Q-learning objectives. This distributional perspective also enables customized policy adaptation, allowing the agent to adjust its behavior toward specific preferences. NV-Dou (Yu et al., 2023) extends Neural Fictitious Self-Play (Heinrich & Silver, 2016) with noisy networks and Q-based policy gradients. Noisy networks combined with noisy buffer sampling enable Peasant players to select high-quality noise for discovering diverse strategies. For policy improvement, it integrates Q-based policy gradients (Mean Actor-Critic) with advantage learning and proximal policy optimization to stabilize training. MDou (Luo et al., 2022) introduces Minimum Split Pruning (MSP) to prune low-probability actions from the action space using heuristics. In addition, it uses a unified network across three positions, simplifying the architecture and enhancing generalization. RARSMSDou (Luo & Tan, 2023) combines PPO, DMC, and a reward shaping strategy based on the minimum number

---

[4]https://www.pagat.com/climbing/doudizhu.html

of card splits to tackle sparse rewards and large action spaces. It further reduces complexity by abstracting actions (from 27,472 to 309) and enhances value estimation by using perfect information in the critic. OADMCDou (Luo et al., 2024) enhances training stability by combining Oracle Guiding with Adaptive Deep Monte Carlo (DMC). It begins with an oracle agent trained using full information, and gradually distills it into a standard agent with partial observability. To mitigate unstable policy updates during training, it employs gradient clipping and update magnitude constraints.

Overall, these works reflect a clear trend toward minimizing reliance on heuristics and human knowl-edge, instead leveraging self-play and scalable sampling-based training. This paradigm shift has substantially improved the efficiency and effectiveness of learning in complex imperfect-information games such as DouDizhu. However, one often overlooked aspect is that most of these methods focus primarily on optimizing reinforcement learning algorithms—such as improving sample effi-ciency—while paying limited attention to the game-theoretical foundations (Zhao et al., 2022; Wang et al., 2022; Yu et al., 2023; Luo et al., 2022; Luo & Tan, 2023; Luo et al., 2024). This is largely due to the empirical success of self-play. In contrast, This paper adopts a fundamentally different perspective by establishing a principled self-play framework with theoretical convergence guarantees alongside strong empirical performance.

# D ANALYSIS OF GENERALIZED WEAKENED FICTITIOUS PLAY AND ITS EQUIVALENT FORM

In Section 2, we briefly reviewed the game-theoretic concepts relevant to our work. In this section, we first provide detailed definitions omitted from the main text for brevity, making the paper self-contained. We then derive the primitive equivalent formulations of the GWFP introduced in Eq. (5).

## D.1 GAME THEORY BACKGROUND AND ITS CONNECTIONS WITH REINFORCEMENT LEARNING

Reinforcement learning (RL) typically models the environment as a Markov Decision Process (MDP), defined by the tuple $(\mathcal{S}, \mathcal{A}, \mathcal{R}, \mathcal{P}, \gamma)$, where $\mathcal{S}$ is the state space, $\mathcal{A}$ is the action space, $\mathcal{R}$ is the reward function, $\mathcal{P}$ is the transition probability function, and $\gamma$ is the discount factor. A key assumption in this framework is the **Markov property**, which means that the future is independent of the past given the current state and action.

In contrast to the Markov property central to MDPs, the analysis of extensive-form games in game theory centers around the concept of an **information set**, denoted $u^i$ for player $i$. An information set represents the collection of game states that are indistinguishable to the player based on their observations.

**On Strategy Representations and the Rationale for Kuhn's Theorem.** Our work connects the classical game-theoretic algorithm of Fictitious Play (FP) with modern deep reinforcement learning (RL). This connection requires careful consideration of how strategies are represented, as the theoretical foundations of FP and the practical implementation of deep RL use different formalisms.

First, the classical theory of FP and its extension GWFP, which provide the convergence guarantees for our method, are defined over the space of mixed strategies. A **mixed strategy** is a probability distribution over all of a player's possible deterministic plans (**pure strategies**). Second, in practice, deep RL agents like FPDou learn behavioral strategies. A **behavioral strategy** specifies action probabilities independently at each information set. This is the natural output of a policy network that processes local information. This creates an apparent mismatch: our algorithm is implemented with behavioral strategies, yet its theoretical justification stems from a framework for mixed strategies. To bridge this gap and ensure our method is theoretically sound, we rely on a cornerstone result from the study of extensive-form games: Kuhn's Theorem (Theorem D.2).

The theorem's validity hinges on the assumption of **perfect recall**. A player has perfect recall if they remember all their own past actions and observations—a condition we assume and approximate in our model by providing sufficient action history to the agent. For a game with perfect recall, Kuhn's Theorem shows that any mixed strategy is **realization-equivalent** to a behavioral strategy. Realization-equivalence (Theorem D.1) means the two strategies are indistinguishable from an opponent's perspective, as they induce the same distribution over game outcomes against the opponents.

The introduction of perfect recall, Kuhn's Theorem, and realization-equivalence is the essential theoretical underpinning of our work. It is precisely what allows us to apply the powerful convergence

results of GWFP to our practical RL-based FPDou agent, formally justifying our use of a behavioral strategy implementation.

**Definition D.1** (**Realization-Equivalence**). Two strategies $\pi^1$ and $\pi^2$ of a player are realization-equivalent if for any fixed strategy profile of the other players both strategies, $\pi^1$ and $\pi^2$, define the same probability distribution over the states of the game.

**Theorem D.2** (**Kuhn's Theorem** (Kuhn, 1953)). *For a player with perfect recall, any mixed strategy is realization-equivalent to a behavioral strategy, and vice versa.*

## D.2 DERIVATION OF THE PRIMITIVE EQUIVALENT FORMULATIONS OF GWFP

As discussed in Section 3.2, the formulation of GWFP given in Definition 3.1 involves two components. Starting from a randomly initialized $\pi_0$: (1) each player $i$ first computes an $\epsilon$-best response $b^i_{\epsilon_t}(\pi_t^{-i})$ against $\pi_t^{-i}$; (2) the average strategy $\pi_{t+1}^i$ is then updated using a weighted combination of the current average strategy $\pi_t^i$ and the $\epsilon$-best response $b^i_{\epsilon_t}(\pi_t^{-i})$:

$$\pi_{t+1}^i = (1 - \alpha_{t+1})\pi_t^i + \alpha_{t+1}b^i_{\epsilon_t}(\pi_t^{-i}). \tag{8}$$

In large-scale games, applying this procedure requires maintaining and updating two separate strategies per player. When implementing these strategies with function approximation, the average strategy $\pi_t^i$ is typically updated via supervised learning, while the best response $b^i_{\epsilon_t}(\pi_t^{-i})$ is learned through reinforcement learning (Heinrich et al., 2015; Heinrich & Silver, 2016). Both learning processes are resource-intensive, involving substantial data and computation. Moreover, the use of separate networks increases the number of parameters and the potential for learning errors.

To reduce complexity and enable a more practical implementation, the two components can be unified into a single update rule using a primitive form of fictitious play. This is feasible because the average strategy $\pi_t^i$ can be expanded as a weighted average of historical best responses. Specifically, we expanded Eq. (8) as an explicit average over the best response sequence:

$$
\begin{aligned}
\pi_{t+1}^i &= (1 - \alpha_{t+1})\pi_t^i + \alpha_{t+1}b^i_{\epsilon_t}(\pi_t^{-i}) \\
&= (1 - \alpha_{t+1})\left((1 - \alpha_t)\pi_{t-1}^i + \alpha_t b^i_{\epsilon_{t-1}}(\pi_{t-1}^{-i})\right) + \alpha_{t+1}b^i_{\epsilon_t}(\pi_t^{-i}) \\
&= (1 - \alpha_{t+1})(1 - \alpha_t)\pi_{t-1}^i + (1 - \alpha_{t+1})\alpha_t b^i_{\epsilon_{t-1}}(\pi_{t-1}^{-i}) + \alpha_{t+1}b^i_{\epsilon_t}(\pi_t^{-i}) \\
&= (1 - \alpha_{t+1})(1 - \alpha_t)\left((1 - \alpha_{t-1})\pi_{t-2}^i + \alpha_{t-1}b^i_{\epsilon_{t-2}}(\pi_{t-2}^{-i})\right) \\
&\quad + (1 - \alpha_{t+1})\alpha_t b^i_{\epsilon_{t-1}}(\pi_{t-1}^{-i}) + \alpha_{t+1}b^i_{\epsilon_t}(\pi_t^{-i}) \\
&= (1 - \alpha_{t+1})(1 - \alpha_t)(1 - \alpha_{t-1})\pi_{t-2}^i + (1 - \alpha_{t+1})(1 - \alpha_t)\alpha_{t-1}b^i_{\epsilon_{t-2}}(\pi_{t-2}^{-i}) \\
&\quad + (1 - \alpha_{t+1})\alpha_t b^i_{\epsilon_{t-1}}(\pi_{t-1}^{-i}) + \alpha_{t+1}b^i_{\epsilon_t}(\pi_t^{-i}) \\
&= \cdots \\
&= \prod_{k=1}^{t+1}(1 - \alpha_k)\pi_0^i + \sum_{k=1}^{t+1}\left(\alpha_k \prod_{j=k+1}^{t+1}(1 - \alpha_j)b^i_{\epsilon_{k-1}}(\pi_{k-1}^{-i})\right) \\
&= \underbrace{\prod_{k=1}^{t+1}(1 - \alpha_k)\pi_0^i + \sum_{k=1}^{t}\left(\alpha_k \prod_{j=k+1}^{t+1}(1 - \alpha_j)b^i_{\epsilon_{k-1}}(\pi_{k-1}^{-i})\right)}_{\text{Past } \epsilon\text{-Best Responses}} + \underbrace{\alpha_{t+1}b^i_{\epsilon_t}(\pi_t^{-i})}_{\text{New } \epsilon\text{-Best Response}}
\end{aligned}
\tag{9}
$$

With $\alpha_t = 1/t$, we have

$$\pi_{t+1}^i = \prod_{k=1}^{t+1}(1-\frac{1}{k})\pi_0^i + \sum_{k=1}^{t+1}\left(\frac{1}{k}\prod_{j=k+1}^{t+1}(1-\frac{1}{j})b_{\epsilon_{k-1}}^i(\pi_{k-1}^{-i})\right)$$

$$= \prod_{k=1}^{t+1}\frac{k-1}{k}\pi_0^i + \sum_{k=1}^{t+1}\left(\frac{1}{k}\prod_{j=k+1}^{t+1}\frac{j-1}{j}b_{\epsilon_{k-1}}^i(\pi_{k-1}^{-i})\right)$$

$$= \frac{1}{2}\cdot\frac{2}{3}\cdots\frac{t}{t+1}\pi_0^i + \sum_{k=1}^{t+1}\left(\frac{1}{k}\cdot\frac{k}{k+1}\cdot\frac{k+1}{k+2}\cdots\frac{t}{t+1}b_{\epsilon_{k-1}}^i(\pi_{k-1}^{-i})\right) \qquad (10)$$

$$= \frac{1}{t+1}\pi_0^i + \sum_{k=1}^{t+1}\left(\frac{1}{t+1}b_{\epsilon_{k-1}}^i(\pi_{k-1}^{-i})\right)$$

$$= \underbrace{\frac{1}{t+1}\pi_0^i + \sum_{k=1}^{t}\left(\frac{1}{t+1}b_{\epsilon_{k-1}}^i(\pi_{k-1}^{-i})\right)}_{\text{Past }\epsilon\text{-Best Responses}} + \underbrace{\frac{1}{t+1}b_{\epsilon_t}^i(\pi_t^{-i})}_{\text{New }\epsilon\text{-Best Response}}$$

Here, our goal is to express the right hand side in a form with consistent semantics, such that we only need to maintain one network to learn $\pi_{t+1}^i$.

Since it is an average of $\epsilon$-best responses, we can use experience replay to consolidate the two-step process in Eq. (8) into a single step, rather than maintaining two separate networks for $\pi_t^i$ and $b_{\epsilon_t}^i(\pi_t^{-i})$ as in Eq. (8). This approach is introduced in our FPDou implementation (see Section 3.3).

Currently, in Eqs. (9) and (10), the right-hand side—aside from $\pi_0$—consists of a sequence of best responses, thus retaining consistent semantics. For $\pi_0^i$, it is a randomly initialized policy whose influence gradually fades as $t$ increases and does not affect convergence, so we do not need to worry about it. Moreover, there is a more reasonably explanation for $\pi_0$: since no prior strategy exists before $\pi_0^i$, $\pi_0^i$ can be viewed as an $\hat{\epsilon}$-best response to an unknown opponent strategy, provided $\hat{\epsilon}$ is set sufficiently large. As specified in Definition 3.1, we only require $\epsilon_t \to 0$ as $t \to \infty$, so the initial value $\hat{\epsilon}$ does not matter. In this way, $\pi_0$ can be treated as an $\epsilon$-best response, making all terms in the right-hand side semantically consistent (i.e., $\epsilon$-best response).

Leveraging this consistency, we can consolidate the two-step process in Eq. (8) into a single step: each player uses one network $\pi_t^i$, eliminating the need to maintain two separate networks for $\pi_t^i$ and $b_{\epsilon_t}^i(\pi_t^{-i})$. Specifically, we split the right-hand side into two term: the first term is the weighted combination of past $\epsilon$-best responses, $\frac{1}{t+1}\pi_0^i + \sum_{k=1}^{t}\left(\frac{1}{t+1}b_{\epsilon_{k-1}}^i(\pi_{k-1}^{-i})\right)$; the second term is a new $\epsilon$-best response, $\frac{1}{t+1}b_{\epsilon_t}^i(\pi_t^{-i})$. By using deep RL to train the new $\epsilon$-best response while simultaneously using supervised learning to learn the weighted combination of past $\epsilon$-best responses, we can directly approximate the next average strategy $\pi_{t+1}^i$. This formulation provides a practical implementation of GWFP for large-scale games, rooted in its theoretical framework.

## E  IMPLEMENTATION DETAILS

### E.1  PSEUDOCODE OF FPDOU

---

**Algorithm 1** FPDou

---

1: Initialize actor networks $f_{\theta_{\text{Landlord}}}, f_{\theta_{\text{Peasant}_1}}, f_{\theta_{\text{Peasant}_2}}$; learner networks $f_{\bar{\theta}_{\text{Landlord}}}, f_{\bar{\theta}_{\text{Peasant}_1}}, f_{\bar{\theta}_{\text{Peasant}_2}}$

2: Initialize win-rate threshold $\tau$, replay buffer $\mathcal{D}$, off-policy data fraction $\lambda = 0.5$, batch size = $B$, on-policy data queue $\mathcal{Q}_{on}$ with capacity $(1 - \lambda)B$, `landlord_on_policy=True, peasants_on_policy=True`

3: **for** iteration $k = 0$ **to** $K$ **do**

4:     `landlord_win_count` $= 0$

5:     **for** game $n = 0$ **to** $N$ **do**

6:         Initialize the environment $s_0 \leftarrow Env$
            /* collect on-policy trajectories */

7:         **while** not terminate **do**

8:             Take actions following `top-k@n` strategy using actor networks $f_{\theta_{\text{Landlord}}}, f_{\theta_{\text{Peasant}_1}}, f_{\theta_{\text{Peasant}_2}}$

9:         **end while**

10:         **if** Landlord won **then**

11:             `landlord_win_count` $+ = 1$

12:         **end if**

13:         Assign final game reward $r_T$ to all state-action pairs $\{(s_i, a_i, r_T, \texttt{position})\}_{i=0}^{T}$, add them to $\mathcal{Q}_{on}$
            /* Sample off-policy data and mix with on-policy data */

14:         **if** $\mathcal{Q}_{on}$ is full **then**

15:             Sample off-policy data with size $\lambda B$ from $\mathcal{D}$, form a batch $\mathcal{B}$ with all data in $\mathcal{Q}_{on}$
                /* update networks */

16:             Update $f_{\theta_{\text{Landlord}}}$ following Eq. (6), update $f_{\theta_{\text{Peasant}_1}}, f_{\theta_{\text{Peasant}_2}}$ following Eq. (7)
                /* synchronize parameters for on-policy learning */

17:             **if** `landlord_on_policy=True` **then**

18:                 $\theta_{\text{Landlord}} \leftarrow \bar{\theta}_{\text{Landlord}}$

19:             **end if**

20:             **if** `peasants_on_policy=True` **then**

21:                 $\theta_{\text{Peasant}_1} \leftarrow \bar{\theta}_{\text{Peasant}_1}, \theta_{\text{Peasant}_2} \leftarrow \bar{\theta}_{\text{Peasant}_2}$

22:             **end if**
                /* store trajectories to buffer after on-policy learning */

23:             Add all data from $\mathcal{Q}_{on}$ to $\mathcal{D}$, clear $\mathcal{Q}_{on}$

24:         **end if**

25:     **end for**
        /* determine on-policy learning for next iteration */

26:     **if** `landlord_win_count`$/N \geq \tau$ **then**

27:         `landlord_on_policy=False,peasants_on_policy=True`

28:     **else**

29:         `landlord_on_policy=True,peasants_on_policy=False`

30:     **end if**

31: **end for**

---

## E.2 CARD REPRESENTATION

We represent both states and actions using a one-hot $4 \times 15$ matrix that encodes the number of cards for each rank, including jokers. Fig. 1 in the main text illustrates a hand card representation. Fig. 5 further show the representation for action `Pass` to make the paper complete. To construct a complete $(s, a)$ representation, we stack the action and state matrices along the channel dimension. The first channel corresponds to the action to be played, while the remaining channels represent the current state. Further details of the card representation are provided in Table 3. Different from previous works, we do not incorporate human designed feature such as the number of bombs (Zha et al., 2021b) and the encoding of the legal combination of cards (Yang et al., 2022). We only stack the card information and the action history, similar to AlphaZero (Silver et al., 2016; 2017; 2018).

| 0 | 0 | 0 | 0 | 0 | 0 | 0 | 0 | 0 | 0 | 0 | 0 | 0 | 1 | 1 |
| 0 | 0 | 0 | 0 | 0 | 0 | 0 | 0 | 0 | 0 | 0 | 0 | 0 | 1 | 1 |
| 0 | 0 | 0 | 0 | 0 | 0 | 0 | 0 | 0 | 0 | 0 | 0 | 0 | 1 | 1 |
| 0 | 0 | 0 | 0 | 0 | 0 | 0 | 0 | 0 | 0 | 0 | 0 | 0 | 0 | 0 |

Figure 5: The representation of action `Pass`.

Table 3: The meaning of each channel in the card representation of the $(s, a)$ pair, Channel 0 represents the action $a$, while other Channels represent the state $s$. Each channel is encoded as a $4 \times 15$ one-hot matrix, and the same representation method is used for all players.

| Channel | Feature |
|---------|---------|
| 0 | Card(s) to be played |
| 1 | The player's current hand cards |
| 2 | Combined hand cards of the other two players |
| 3 | Same as Channel 2[5] |
| 4 | Cards played by the player |
| 5 | Cards played by the previous player |
| 6 | Cards played by the next player |
| 7 | The three bidding cards |
| 8-68 | Action history over the last 60 steps |

---

[5]Channel 2 and 3 are identical because we cannot distinguish the other players' hands. However, when we apply regularization for the Peasants, these channels will be separated, with each representing one of the other two players' hands.

## E.3 NETWORK ARCHITECTURE

We maintain three networks for the three positions, following previous works (Zha et al., 2021b; Yang et al., 2022). Each network shares the same architecture: a distributional Q-network (Bellemare et al., 2017), implemented as a convolutional neural network (CNN) with skip connections (He et al., 2016). The input is a stack of the current state and a legal action, representing the $(s, a)$ pair. The output is a distribution over Q-values for action $a$ given state $s$. We use 8 bins to represent different outcomes: winning or losing with 0, 1, 2, or 3 or more bombs, where more bombs correspond to higher scores. Details of the network architecture are provided in Tables 4 and 5.

Table 4: Structure of our Q-network.

| Layer | Configuration |
|---|---|
| Conv1 | kernel_size=1, stride=1, padding=2, out_channels=64 |
| ResBlock 1-5 | See Table 5 |
| Conv2 | kernel_size=1, stride=1, out_channels=2 |
| BatchNorm | - |
| Flatten | - |
| MLP | feature_dim=304, action_dim=8 |
| Softmax | - |

Table 5: Structure of the ResBlock used in our Q-network.

| Layer | Configuration | Activation |
|---|---|---|
| Conv1 | in_channels=64, kernel_size=3, stride=1, padding=1, bias=False, out_channels=64 | None |
| BatchNorm1 | - | ReLU |
| Conv2 | in_channels=64, kernel_size=3, stride=1, padding=1, bias=False, out_channels=64 | None |
| BatchNorm2 | - | None |
| Shortcut | Identity or $1 \times 1$ conv (if shape mismatch) | - |
| Add | - | ReLU |

### E.4 HYPERPARAMETERS

Our experiments were conducted on a server equipped with two AMD EPYC 7313 CPUs (32 cores, 32 threads) and six NVIDIA GPUs (one RTX A5000, two TITAN RTX, and three RTX 2080 Ti). One GPU was used for training, while the remaining five were allocated for simulation. The model was optimized using the AdamW optimizer (Kingma & Ba, 2014; Loshchilov & Hutter, 2019), with a batch size of 2048 and a learning rate of 0.001. Training was performed over a period of one month. The full set of hyperparameters is listed in Table 6.

Table 6: Hyperparameters used in FPDou.

| Category | Hyperparameter | Value |
|---|---|---|
| Network | Input size | $4 \times 15 \times 69$ |
| | Semantic meaning of input channels | See Table 3 |
| | Network architecture | See Tables 4 and 5 |
| | Output size | 8 |
| Exploration | Method | top-k@n (k=3,n=3) |
| | Actions k | 3 |
| | Steps n | 3 |
| Training | Batch size | 2048 |
| | Learning rate | 0.001 |
| | Optimizer | AdamW |
| | Peasants regularization parameter | 0.01 |
| | Fraction of off-policy data in each batch | 0.5 |
| | Buffer size | 100,000 |
| | Winning threshold | 0.5 |
| | Evaluation window size | 200 |
| | Actor GPU number | 5 |
| | Learner GPU number | 1 |
| | Number of actors per GPU | 2 |

## F APPROXIMATIONS OF FPDOU RELATIVE TO GWFP'S THEORETICAL CONDITIONS

Our FPDou method is rooted in the theory of GWFP for solving the complex DouDizhu game. While we strive to align FPDou with GWFP's theoretical convergence conditions, practical constraints complicate this effort. These include the game's inherent complexity and the use of neural networks for function approximation, leading to unavoidable gaps and necessitating mild simplifying assumptions. This section first details the convergence conditions of GWFP, then systematically discusses how FPDou is implemented to satisfy these conditions, along with the assumptions and approximations underlying our design.

Per Definition 3.1, GWFP's convergence relies on specific conditions for the step size $\alpha_t$, step size summation $\sum_{t=1}^{\infty} \alpha_t$, best response margin $\epsilon_t$, and perturbation term $M_t$. Additionally, since FPDou learns behavioral strategies rather than mixed strategies, it must satisfy the realization-equivalence condition defined in Definition D.1, a requirement for perfect recall. Collectively, the GWFP convergence conditions are:

- $\alpha_t \to 0$ as $t \to \infty$

- $\sum_{t=1}^{\infty} \alpha_t = \infty$ as $t \to \infty$

- $\epsilon_t \to 0$ as $t \to \infty$

- $\forall\, T > 0,\ \lim_{t \to \infty} \sup_k \left\{ \left\| \sum_{i=t}^{k-1} \alpha_{i+1} M_{i+1} \right\| \text{ s.t. } \sum_{i=t}^{k-1} \alpha_{i+1} \leq T \right\} = 0.$

- Perfect recall: each player remembers the full history $\{u_1^i, a_1^i, \cdots, u_k^i\}$ leading to the current information set $u_k^i$.

**Step Size** $\alpha_t$**.** FPDou's implementation adheres to the step size schedule defined in Eq. (10). With a sufficiently large replay buffer, $\alpha_t$ can be approximately regarded as $1/t$. Under this approximation, it naturally satisfies $\alpha_t \to 0$ as $t \to \infty$.

**Step Size Summation** $\sum_{t=1}^{\infty} \alpha_t$**.** For $\alpha_t = 1/t$, the summation $\sum_{t=1}^{\infty} \alpha_t = \sum_{t=1}^{\infty} \frac{1}{t}$ (the harmonic series) is known to diverge to infinity. This divergence can be rigorously verified via term grouping:

$$\sum_{t=1}^{\infty} \frac{1}{t} = 1 + \frac{1}{2} + (\frac{1}{3} + \frac{1}{4}) + (\frac{1}{5} + \frac{1}{6} + \frac{1}{7} + \frac{1}{8}) + \cdots \qquad (11)$$

Each grouped term is lower-bounded by $\frac{1}{2}$ (e.g., $\frac{1}{3} + \frac{1}{4} > \frac{1}{4} + \frac{1}{4} = \frac{1}{2}$), The summation thus reduces to $1 + \frac{1}{2} + \frac{1}{2} + \frac{1}{2} + \cdots$, which clearly diverges to infinity. This confirms FPDou satisfies GWFP's step size summation condition.

**Best Response Margin** $\epsilon_t$**.** In FPDou's $\epsilon$-best response learning loop, we use a win-rate threshold $\tau = 0.5$ to determine whether the current training qualifies as an $\epsilon_t$-best response policy. Early in training, opponent policies are relatively weak, so achieving a 0.5 win rate corresponds to an $\epsilon_t$-best response with $\epsilon_t > 0$ (i.e., the policy is suboptimal but competitive). As training progresses, all players improve. Attaining a win rate of 0.5 corresponds to a decreasing $\epsilon_t$. To formalize the convergence $\epsilon_t \to 0$, we introduce a mild, empirically motivated assumption: DouDizhu is a balanced game, where a 0.5 win rate aligns with the performance of a Nash equilibrium policy. Under this assumption, the monotonic decrease of $\epsilon_t$ implies $\epsilon_t \to 0$ as training converges.

Noting that perfect game balance cannot be guaranteed in practice (given the unknown true Nash equilibrium of DouDizhu), we conducted a supplementary ablation study using an adaptive win-rate threshold for reference. Detailed results of this study are provided in Section G.5.

**Perturbation Term** $M_t$**.** We want to prove that the perturbation condition required by GWFP is satisfied. The condition is given by:

$$\lim_{t \to \infty} \sup_k \left\{ \left\| \sum_{i=t}^{k-1} \alpha_{i+1} M_{i+1} \right\| \text{ s.t. } \sum_{i=t}^{k-1} \alpha_{i+1} \leq T \right\} = 0,$$

given $\alpha_t = 1/t$ and a sequence of perturbations $\{M_t\}$. To apply the GWFP framework to our method, we define the perturbation term $M_t$ as the neural network's approximation error $Q - \widehat{Q}$: the difference between the true value ($Q$) and the predicted value ($\widehat{Q}$). Empirically, as shown in Fig. 6, this term is bounded and exhibits zero-mean stochastic oscillation. Based on this observation and the standard context of stochastic training methods, we establish the following reasonable assumptions:

1. **Boundedness:** The sequence of perturbations $\{M_t\}$ is bounded. Fig. 6 shows that after an initial transient phase, the values remain within a fixed range. Thus, there exists a constant $C > 0$ such that $\|M_t\| \leq C$ for all $t$.

2. **Zero Mean:** The perturbations oscillate around zero, consistent with unbiased stochastic noise. We assume the perturbations have a zero mean: $\mathbb{E}[M_t] = 0$ for all $t$.

3. **Uncorrelation:** The noise generated by stochastic gradient descent at different iterations is typically assumed to be uncorrelated. Thus, we assume $\mathbb{E}[M_i M_j] = 0$ for all $i \neq j$.

*Proof.* Let $S_{t,k}$ denote the partial sum of the perturbations:

$$S_{t,k} = \sum_{i=t}^{k-1} \alpha_{i+1} M_{i+1}$$

Our goal is to show that $S_{t,k}$ converges to zero in a manner that satisfies the condition. We will prove this by showing that $S_{t,k}$ converges in mean square to zero, which implies convergence in probability.

First, let us compute the expectation of $S_{t,k}$. By the linearity of expectation and Assumption 2 (Zero Mean):

$$\mathbb{E}[S_{t,k}] = \mathbb{E}\left[\sum_{i=t}^{k-1} \alpha_{i+1} M_{i+1}\right]$$

$$= \sum_{i=t}^{k-1} \alpha_{i+1} \mathbb{E}[M_{i+1}]$$

$$= \sum_{i=t}^{k-1} \alpha_{i+1} \cdot 0 = 0.$$

Next, we analyze the second moment, $\mathbb{E}[\|S_{t,k}\|^2]$. Since the mean is zero, this is equivalent to the trace of the covariance matrix.

$$\mathbb{E}[\|S_{t,k}\|^2] = \mathbb{E}\left[\left(\sum_{i=t}^{k-1} \alpha_{i+1} M_{i+1}\right)\left(\sum_{j=t}^{k-1} \alpha_{j+1} M_{j+1}\right)\right]$$

$$= \mathbb{E}\left[\sum_{i=t}^{k-1}\sum_{j=t}^{k-1} \alpha_{i+1}\alpha_{j+1} M_{i+1} M_{j+1}\right]$$

$$= \sum_{i=t}^{k-1}\sum_{j=t}^{k-1} \alpha_{i+1}\alpha_{j+1} \mathbb{E}[M_{i+1} M_{j+1}].$$

By Assumption 3 (Uncorrelation), the cross-terms where $i \neq j$ vanish. The sum simplifies to the terms where $i = j$:

$$\mathbb{E}[\|S_{t,k}\|^2] = \sum_{i=t}^{k-1} \alpha_{i+1}^2 \mathbb{E}[M_{i+1} M_{i+1}]$$

$$= \sum_{i=t}^{k-1} \alpha_{i+1}^2 \mathbb{E}[\|M_{i+1}\|^2].$$

Using Assumption 1 (Boundedness), we know that $\|M_{i+1}\| \leq C$, which implies $\|M_{i+1}\|^2 \leq C^2$. Therefore, $\mathbb{E}[\|M_{i+1}\|^2] \leq C^2$. We can now bound the expression:

$$\mathbb{E}[\|S_{t,k}\|^2] \leq \sum_{i=t}^{k-1} \alpha_{i+1}^2 C^2.$$

Substituting $\alpha_t = 1/t$, we have $\alpha_{i+1} = 1/(i+1)$:

$$\mathbb{E}[\|S_{t,k}\|^2] \leq C^2 \sum_{i=t}^{k-1} \frac{1}{(i+1)^2}.$$

To find an upper bound that is independent of $k$, we extend the sum to infinity:

$$\mathbb{E}[\|S_{t,k}\|^2] \leq C^2 \sum_{i=t}^{\infty} \frac{1}{(i+1)^2}.$$

The series $\sum_{n=1}^{\infty} \frac{1}{n^2}$ is a convergent p-series (with $p = 2 > 1$). A fundamental property of a convergent series is that its tail must converge to zero. Therefore:

$$\lim_{t\to\infty} \sum_{i=t}^{\infty} \frac{1}{(i+1)^2} = 0.$$

This implies that the upper bound on $\mathbb{E}[\|S_{t,k}\|^2]$ converges to zero as $t \to \infty$, uniformly for all $k$:

$$\lim_{t\to\infty} \sup_k \mathbb{E}[\|S_{t,k}\|^2] \leq \lim_{t\to\infty} C^2 \sum_{i=t}^{\infty} \frac{1}{(i+1)^2} = 0.$$

Since $\mathbb{E}[\|S_{t,k}\|^2] \geq 0$, by the Squeeze Theorem, we have shown that $S_{t,k}$ converges in mean square to zero:

$$\lim_{t\to\infty} \sup_k \mathbb{E}[\|S_{t,k}\|^2] = 0.$$

Convergence in mean square implies convergence in probability. For any $\epsilon > 0$, by Chebyshev's inequality:

$$P(\|S_{t,k}\| \geq \epsilon) \leq \frac{\mathbb{E}[\|S_{t,k}\|^2]}{\epsilon^2}.$$

Taking the limit as $t \to \infty$ and the supremum over $k$:

$$\lim_{t\to\infty} \sup_k P(\|S_{t,k}\| \geq \epsilon) \leq \lim_{t\to\infty} \sup_k \frac{\mathbb{E}[\|S_{t,k}\|^2]}{\epsilon^2} = 0.$$

This confirms that $\|S_{t,k}\|$ converges to 0 in probability, which is sufficient to satisfy the GWFP condition. Thus, the perturbation condition is met. $\square$

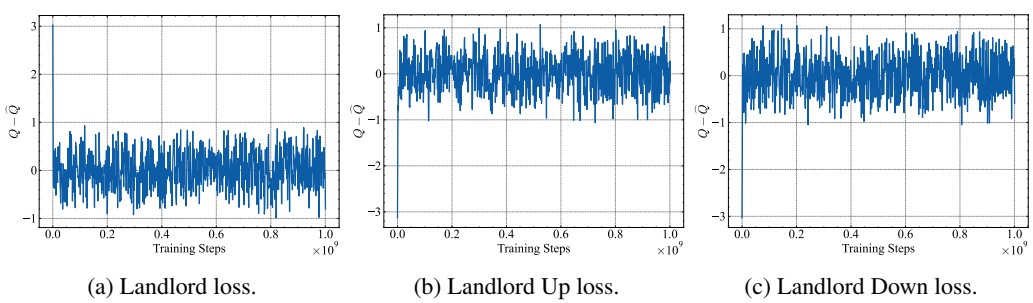

| (a) Landlord loss. | (b) Landlord Up loss. | (c) Landlord Down loss. |

Figure 6: Empirical behavior of the perturbation term $M_t = Q - \widehat{Q}$. After an initial phase, the perturbation is bounded and exhibits zero-mean stochastic oscillations. This provides experimental validation for the core assumptions regarding the perturbation term in our GWFP convergence analysis.

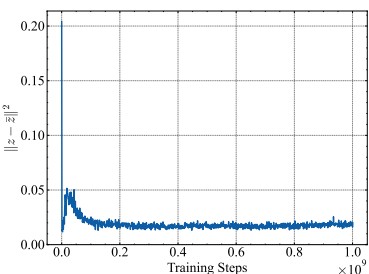

Figure 7: The L2 loss between latent representations from imperfect ($z$) and perfect ($\bar{z}$) information decreases rapidly and keeps at a low value. This empirically validates our approach, confirming that the network successfully learns to bridge the training-execution gap by aligning its internal representations.

**Perfect Recall.** Perfect recall requires full history access, which is approximated in FPDou by stacking the current hand information with the previous 60 actions in our representation. As shown in Table 7, 99% of DouDizhu games are completed within 60 steps, ensuring this truncated history captures nearly all relevant context for decision-making. This approximation is sufficiently accurate to maintain realization equivalence in practice.

From the above discussion, while FPDou is designed to align closely with GWFP theory, there are still some unavoidable approximations arise from practical constraints. (1) Finite Replay Buffer: GWFP theoretically requires an infinite buffer to store all training experience, which is infeasible. Thus, FPDou instead uses a large buffer that our server can maintain, which suffices for stable training. (2) Balanced Game Assumption: Without knowledge of DouDizhu's true Nash equilibrium, we assume a 0.5 win rate approximates equilibrium performance. This may affect the final performance as we cannot guarantee converge to a Nash equilibrium. But as we discussed, we can at least

ensures $\epsilon_t$ decreases during the learning, and we got a strong empirical performance though with this assumption. (3) Truncated History for Perfect Recall: Stacking 60 historical actions captures 99% of game histories, but not all edge cases. This mild approximation can be refined by increasing the stack length if computational resources are available.

Overall, FPDou is intentionally designed to minimize deviations from GWFP's theoretically rigorous framework, with approximations limited to practical feasibility. Empirical results validate that grounding algorithm design in GWFP theory yields significant performance advantages, supporting the reasonableness of our approach and its approximations.

# G   ADDITIONAL RESULTS

Our method is grounded in the principles of Generalized Weakened Fictitious Play (GWFP) (Heinrich et al., 2015), which provides convergence guarantees. Several parameters are chosen based on game statistics to support the algorithm's convergence. In this section, we first present key characteristics of the game during training, such as game length and the distribution of game outcomes. We then provide additional experimental results to demonstrate the performance of FPDou compared to existing DouDizhu programs.

## G.1   GAME STATISTICS OBSERVED DURING TRAINING

**Game Length.** We measure the game length percentiles over the first $1 \times 10^9$ environment steps. As shown in Table 7, 99% of the games are shorter than 60 steps, with a median length of 33.96. This is significantly shorter than other games such as Go (Silver et al., 2016) and Atari (Bellemare et al., 2013), which can involve hundreds or even thousands of steps. Based on this observation, we stack the last 60 actions in the state representation, using zero padding if necessary, to approximate perfect recall.

Table 7: Game length percentiles. 99% of the games are completed within 60 steps.

| Percentile (%) | 50 | 75 | 90 | 95 | 99 |
|---|---|---|---|---|---|
| Game Length | 33.96 | 41.13 | 47.97 | 51.96 | 60.12 |

**Distribution of Game Outcomes.** We analyze the distribution of game results over the first $1 \times 10^9$ environment steps. The scoring system in DouDizhu starts with a base score of 1, which is multiplied by a dynamic factor that increases with each bomb (including rockets) played. For example, if one bomb is played during a game, the final score is multiplied by 2; if two bombs are played, it is multiplied by 3, and so on. There are no draws in DouDizhu, each game results in a clear win or loss.

As shown in Fig. 8, most games end with 0 or 1 bomb, while outcomes involving three or more bombs are rare, accounting for fewer than 0.5% of all games. Based on this observation, we categorize outcomes into 8 bins: win or loss with 0, 1, 2, or 3 or more bombs. This binning allows us to capture the essential characteristics of the return distribution while maintaining a compact representation.

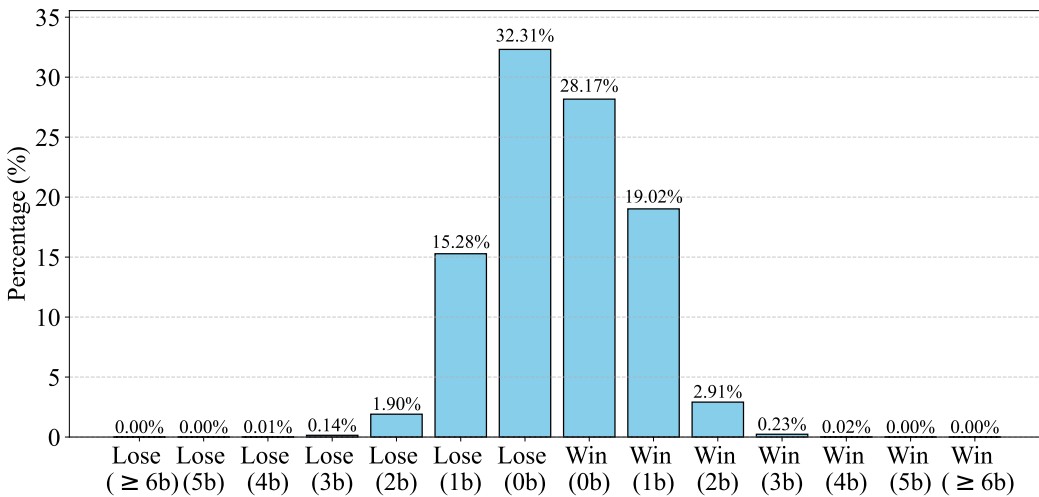

Figure 8: Distribution of game outcomes from the Landlord's perspective. The x-axis indicates the number of bombs played, and the y-axis shows the proportion of games ending with that count. Most games conclude with 0 or 1 bomb, while outcomes with 3 or more bombs are rare, accounting for less than 0.5% of all games.

## G.2 PERFORMANCE COMPARISON WITH EXISTING DOUDIZHU PROGRAMS

We report FPDou's average performance in the main page. This section provides full results. Table 8 shows the unrounded final performance averaged over Landlord and Peasants. Tables 9 and 10 present separate WP and ADP results for each side. Figures 9 and 10 depict the win rate and score learning curves of FPDou versus baseline methods.

Table 8: Performance of FPDou against baselines over 10,000 decks (not rounded). A outperforms B if WP> 0.5 or ADP> 0 (in boldface). Methods are ranked based on ADP.

| Rank | B \ A | FPDou | | PerfectDou | | DouZero | | DouZero (WP) | | SL | | RLCard | | Random | |
|---|---|---|---|---|---|---|---|---|---|---|---|---|---|---|---|
| | | WP | ADP | WP | ADP | WP | ADP | WP | ADP | WP | ADP | WP | ADP | WP | ADP |
| 1 | FPDou | - | - | **0.5201** | **0.0997** | **0.5615** | **0.1974** | **0.5103** | **0.3334** | **0.6837** | **0.9963** | **0.89395** | **2.5218** | **0.99335** | **3.1071** |
| 2 | PerfectDou | 0.4799 | -0.0997 | - | - | **0.54275** | **0.1405** | 0.48935 | **0.2123** | **0.6692** | **1.0329** | **0.8898** | **2.4951** | **0.99275** | **3.0867** |
| 3 | DouZero | 0.4385 | -0.1974 | 0.45725 | -0.1405 | - | - | 0.4525 | **0.1192** | **0.6111** | **0.7739** | **0.8565** | **2.3774** | **0.98675** | **3.0432** |
| 4 | DouZero (WP) | 0.4897 | -0.3334 | **0.51065** | -0.2123 | **0.5475** | -0.1192 | - | - | **0.6596** | **0.7147** | **0.88415** | **2.164** | **0.9881** | **2.7409** |
| 5 | SL | 0.3163 | -0.9963 | 0.3308 | -1.0329 | 0.3889 | -0.7739 | 0.3404 | -0.7147 | - | - | **0.80835** | **1.787** | **0.97385** | **2.6957** |
| 6 | RLCard | 0.10605 | -2.5218 | 0.1102 | -2.4951 | 0.1435 | -2.3774 | 0.11585 | -2.164 | 0.19165 | -1.787 | - | - | **0.9419** | **2.5043** |
| 7 | Random | 0.00665 | -3.1071 | 0.00725 | -3.0867 | 0.01325 | -3.0432 | 0.0119 | -2.7409 | 0.02615 | -2.6957 | 0.0581 | -2.5043 | - | - |

Table 9: WP performance of FPDou against baselines over 10,000 decks (not rounded). A outperforms B if WP> 0.5 (in boldface).

| Rank | B \ A | FPDou | | DouZero (WP) | | PerfectDou | | DouZero | | SL | | RLCard | | Random | |
|---|---|---|---|---|---|---|---|---|---|---|---|---|---|---|---|
| | | L | P | L | P | L | P | L | P | L | P | L | P | L | P |
| 1 | FPDou | 0.4062 | **0.5938** | **0.4202** | **0.6004** | **0.4199** | **0.6203** | 0.4859 | **0.6371** | **0.6018** | **0.7656** | **0.8809** | **0.907** | **0.991** | **0.9957** |
| 2 | DouZero (WP) | 0.3996 | **0.5798** | 0.4147 | **0.5853** | **0.4103** | **0.611** | **0.4698** | **0.6252** | **0.564** | **0.7552** | **0.8737** | **0.8946** | **0.9829** | **0.9933** |
| 3 | PerfectDou | 0.3797 | **0.5801** | 0.389 | **0.5897** | 0.3922 | **0.6078** | **0.4533** | **0.6322** | **0.57** | **0.7684** | **0.8695** | **0.9101** | **0.9882** | **0.9973** |
| 4 | DouZero | 0.3629 | **0.5141** | 0.3748 | **0.5302** | 0.3678 | **0.5467** | 0.4266 | **0.5734** | **0.5223** | **0.6999** | **0.8483** | **0.8647** | **0.9817** | **0.9918** |
| 5 | SL | 0.2344 | 0.3982 | 0.2448 | 0.436 | 0.2316 | 0.43 | 0.3001 | 0.4777 | 0.4066 | **0.5934** | **0.7645** | **0.8522** | **0.9552** | **0.9925** |
| 6 | RLCard | 0.093 | 0.1191 | 0.1054 | 0.1263 | 0.0899 | 0.1305 | 0.1353 | 0.1517 | 0.1478 | 0.2355 | 0.465 | **0.535** | **0.9299** | **0.9539** |
| 7 | Random | 0.0043 | 0.009 | 0.0067 | 0.0171 | 0.0027 | 0.0118 | 0.0082 | 0.0183 | 0.0075 | 0.0448 | 0.0461 | 0.0701 | 0.3593 | **0.6407** |

Table 10: ADP performance of FPDou against baselines over 10,000 decks (not rounded). `A` outperforms `B` if ADP> 0 (in boldface).

| Rank | B \ A | FPDou | | PerfectDou | | DouZero | | DouZero (WP) | | SL | | RLCard | | Random | |
|------|-------|-------|-------|-----------|-------|---------|-------|-------------|-------|-------|-------|--------|-------|--------|-------|
| | | L | P | L | P | L | P | L | P | L | P | L | P | L | P |
| 1 | FPDou | -0.5254 | 0.5254 | **-0.453** | **0.6524** | **-0.2532** | **0.648** | **-0.1482** | **0.815** | 0.5458 | 1.4468 | 2.4084 | 2.6352 | 3.2358 | 2.9784 |
| 2 | PerfectDou | -0.6524 | 0.453 | -0.5532 | 0.5532 | **-0.3476** | **0.6286** | **-0.2992** | **0.7238** | 0.5396 | 1.5262 | 2.3498 | 2.6404 | 3.251 | 2.9224 |
| 3 | DouZero | -0.648 | 0.2532 | -0.6286 | 0.3476 | -0.4294 | 0.4294 | **-0.2912** | **0.5296** | 0.3588 | 1.189 | 2.3222 | 2.4326 | 3.2562 | 2.8302 |
| 4 | DouZero (WP) | -0.815 | 0.1482 | -0.7238 | 0.2992 | -0.5296 | 0.2912 | -0.4116 | 0.4116 | 0.2908 | 1.1386 | 2.1086 | 2.2194 | 2.9314 | 2.5504 |
| 5 | SL | -1.4468 | -0.5458 | -1.5262 | -0.5396 | -1.189 | -0.3588 | -1.1386 | -0.2908 | -0.3784 | 0.3784 | **1.6732** | **1.9008** | 2.9762 | 2.4152 |
| 6 | RLCard | -2.6352 | -2.4084 | -2.6404 | -2.3498 | -2.4326 | -2.3222 | -2.2194 | -2.1086 | -1.9008 | -1.6732 | -0.1772 | 0.1772 | **2.6864** | **2.3222** |
| 7 | Random | -2.9784 | -3.2358 | -2.9224 | -3.251 | -2.8302 | -3.2562 | -2.5504 | -2.9314 | -2.4152 | -2.9762 | -2.3222 | -2.6864 | -0.8398 | 0.8398 |

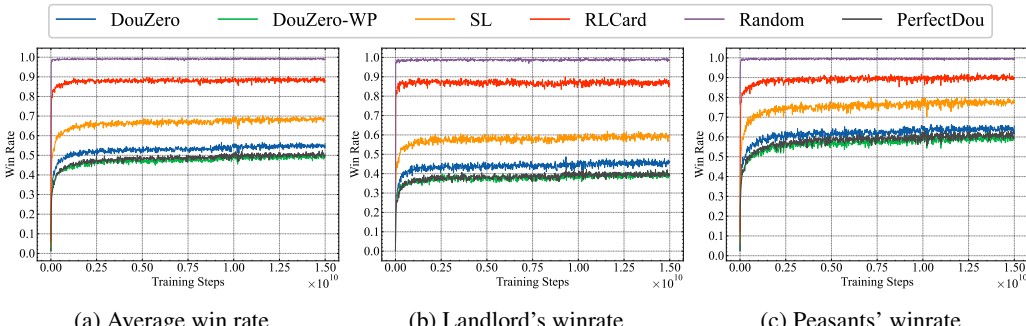

(a) Average win rate       (b) Landlord's winrate       (c) Peasants' winrate

Figure 9: Win rate (WP) learning curves of FPDou against baseline methods.

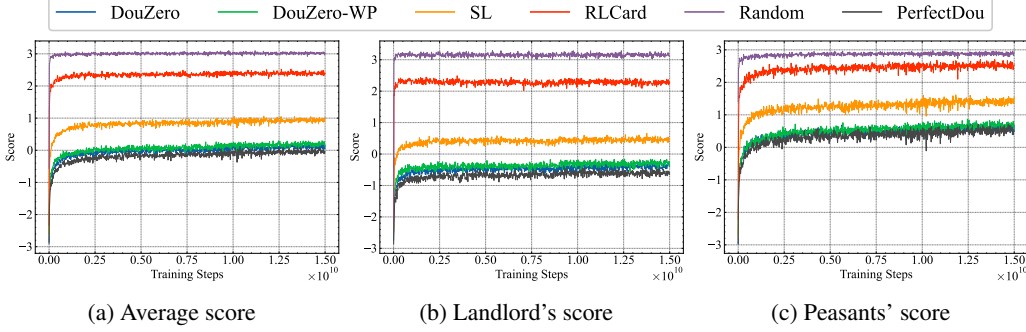

(a) Average score       (b) Landlord's score       (c) Peasants' score

Figure 10: Score (ADP) learning curves of FPDou against baseline methods.

### G.3 RESULTS ON THE BOTZONE PLATFORM

Botzone [6] is a universal online multi-agent game AI platform, designed to evaluate different implementations of game AI by applying them to agents (Bot) and compete with each other. The platform currently supports 33 distinct games, including DouDizhu (FightTheLandlord [7]). To further validate FPDou's practical performance, we upload FPDou to botzone and compete with other bots. As of the experimental cutoff, Botzone hosts 452 independent DouDizhu Bots. As illustrated in Fig. 11, FPDou achieved the top rank among all these competing Bots.

However, the platform's constraints introduce instability to the ranking: first, the daily match volume is limited (with one game completed every 30 minutes), leading to insufficient statistical samples; second, its scoring rules diverge from the evaluation settings in our main paper. As a result, FPDou's rank on Botzone fluctuates: it has maintained the first position for consecutive days (with a score of 1600–1650) but also oscillated to the 10–20th range (with a score of 1500–1550). To avoid presenting unrepresentative snapshot results, we do not include this platform-based evaluation in the main text. Instead, we provide it in the Appendix as supplementary evidence. This not only confirms FPDou's capability to reach the top rank but also offers additional support for its strong practical performance.

To further validate the performance of FPDou, we initiated competitions on Botzone against the top 30-ranked agents. FPDou competes with each agent for 512 decks, where each deck is played twice: method A first plays as Landlord and method B as Peasants, then they switch roles and replay the same deck. As shown in Table 11, FPDou achieves a Winning Percentage (WP) exceeding 0.5 against all opponents, indicating it outperforms all these top bots on the Botzone platform. Notably, FPdou defeats most of the top 30 agents by a substantial margin, with clear advantages ranging from over 55% WP to more than 70%. These results further confirm that FPDou reaches a new state-of-the-art among both open-source and closed-source DouDizhu bots. The agents closest to FPDou in performance are Douzero-ResNet (50.293%) and AI-Doudizhu (50.723%). Notably, our method uses a considerably smaller model while focusing on optimizing the game framework—an approach orthogonal to strategies like modifying RL techniques, scaling network parameters, or refining code with C++ for faster training. These directions could thus serve as promising future work to further enhance FPDou's performance.

---

[6] https://botzone.org.cn/
[7] https://botzone.org.cn/game/FightTheLandlord

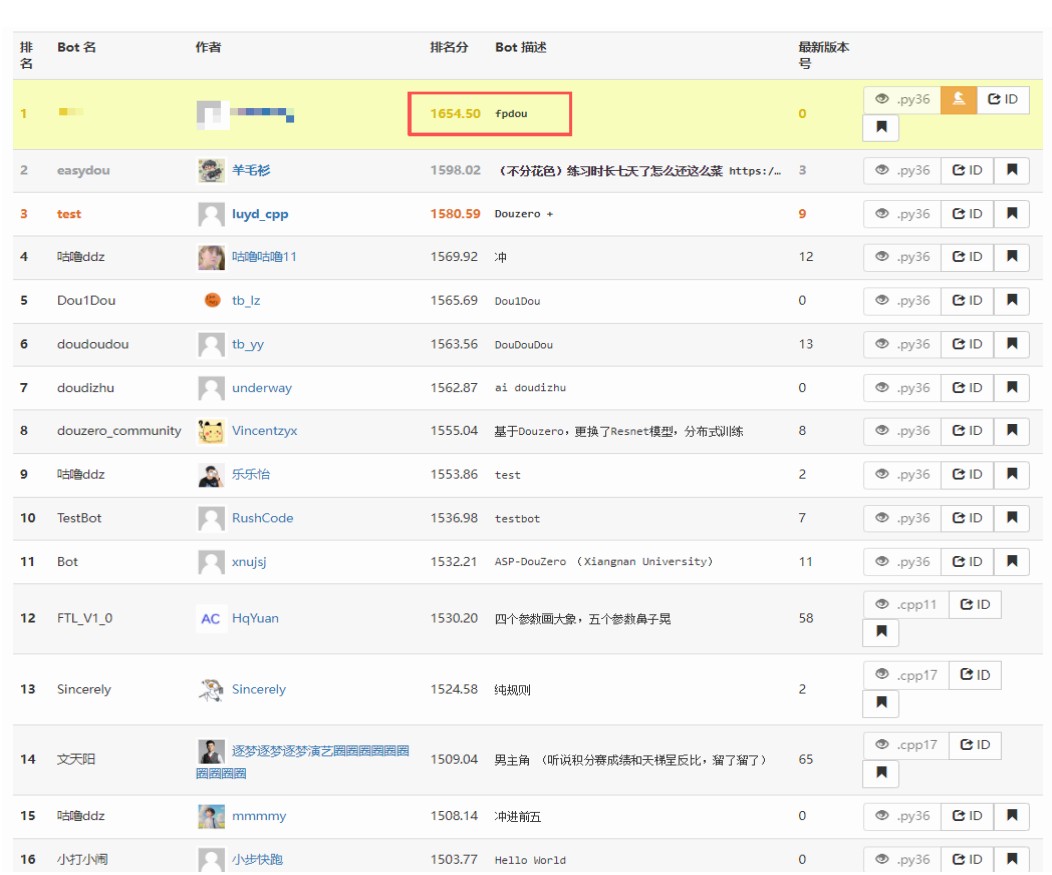

Figure 11: A snapshot of FPDou's performance on Botzone platform. FPDou ranks first among 452 bots on the platform.

Table 11: Performance comparison of FPDou against the top 30 AI agents on the Botzone platform. FPDou achieves a Winning Percentage (WP) exceeding 0.5 against all opponents, indicating it outperforms these top bots and reaches a new state-of-the-art performance among both open-source and closed-source DouDizhu bots.

| Bot Name | Bot Description | Peasants WP(%) | Landlord WP(%) | Average WP(%) |
| --- | --- | --- | --- | --- |
| douzero_community | Douzero-ResNet | 61.523 | 39.062 | 50.293 |
| easydou | AI-Doudizhu | 62.031 | 39.414 | 50.723 |
| dou | | 62.109 | 40.820 | 51.465 |
| biubiubiu | DouZero | 62.187 | 40.742 | 51.465 |
| 咕噜 ddz | | 64.844 | 39.258 | 52.051 |
| doudoudou | | 61.914 | 42.578 | 52.246 |
| Dou1Dou | | 60.938 | 44.727 | 52.832 |
| TestBot | | 63.867 | 41.992 | 52.930 |
| 小打小闹 | | 64.648 | 42.969 | 53.809 |
| test | Douzero+ | 66.406 | 45.117 | 55.762 |
| 咕噜 ddz | | 66.016 | 46.875 | 56.445 |
| luckinluck | Vanilla-DouZero | 65.430 | 48.242 | 56.836 |
| Bot | ASP-DouZero | 66.211 | 48.828 | 57.520 |
| 三傻斗地主 | | 71.875 | 48.438 | 60.156 |
| ddz | | 67.578 | 56.055 | 61.816 |
| ddz | | 67.773 | 59.766 | 63.770 |
| ddz | | 67.773 | 62.891 | 65.332 |
| 这是真的 sample | | 75.391 | 60.547 | 67.969 |
| 反冲机 | | 77.734 | 59.766 | 68.750 |
| 人工 ZZ | | 75.781 | 61.719 | 68.750 |
| FTL_V1_0 | | 73.438 | 65.039 | 69.238 |
| nobody_knows_why | | 80.469 | 58.789 | 69.629 |
| 文天阳 | | 79.492 | 61.133 | 70.312 |
| testbot | | 79.297 | 61.328 | 70.312 |
| 下个 Bot 见 | | 79.883 | 63.867 | 71.875 |
| sample | | 82.227 | 61.914 | 72.070 |
| 写 bug 到凌晨 | | 77.930 | 67.383 | 72.656 |
| 小 ** 熊 | | 81.055 | 66.602 | 73.828 |
| 对三要不起 | | 81.641 | 66.016 | 73.828 |
| 知世就是力量 | | 79.688 | 69.336 | 74.512 |

*†Notes*: Vanilla-DouZero indicates the models from Zha et al. (2021b), while Douzero+ is sourced from Zhao et al. (2022). Douzero-ResNet[a] and AI-Doudizhu[b] are variants based on their respective official descriptions and publicly available implementations.
[a] https://github.com/Vincentzyx/Douzero_Resnet
[b] https://github.com/MingshiYangUIUC/AI-Doudizhu

### G.4 MORE ABLATION STUDY RESULTS

We present additional ablation results beyond those in the main page. Specifically, we compare the impact of different exploration strategies, Q-network depths (number of residual blocks), and batch sizes, using both the SL model and PerfectDou as baselines. The results are shown in Fig. 12 and Fig. 13. Consistent with the findings in the main paper, we observe that a greedy policy slightly outperforms $\epsilon$-greedy, while top-$k$ and softmax@$n$ strategies perform poorly, suggesting that excessive exploration is unnecessary. Additionally, larger models and batch sizes consistently lead to better performance. In addition, we provide a comparison of training allocation and winning percentages by side with smoothing for reference in Fig. 14.

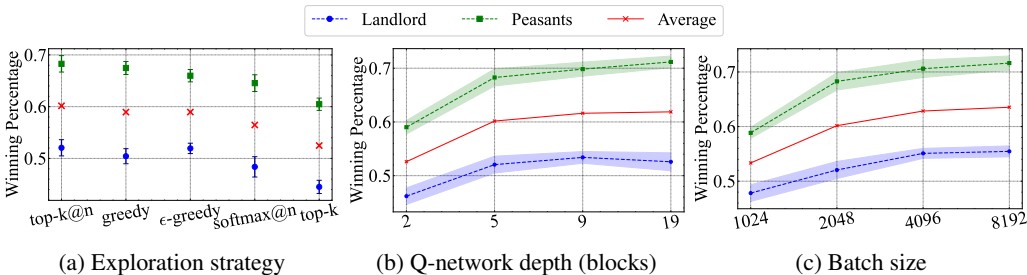

|     |     |     |
| --- | --- | --- |
| (a) Exploration strategy | (b) Q-network depth (blocks) | (c) Batch size |

Figure 12: Winning percentage against SL model under different settings. (a) Greedy policy slightly outperforms $\epsilon$-greedy, while top-k and softmax@n perform poorly, indicating excessive exploration is unnecessary. (b,c) Larger models and batch sizes improve performance.

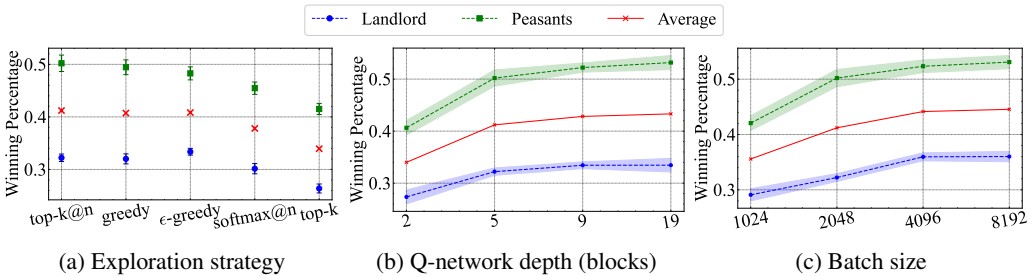

|     |     |     |
| --- | --- | --- |
| (a) Exploration strategy | (b) Q-network depth (blocks) | (c) Batch size |

Figure 13: Winning percentage against PerfectDou under different settings. (a) Greedy policy slightly outperforms $\epsilon$-greedy, while top-k and softmax@n perform poorly, indicating excessive exploration is unnecessary. (b,c) Larger models and batch sizes improve performance.

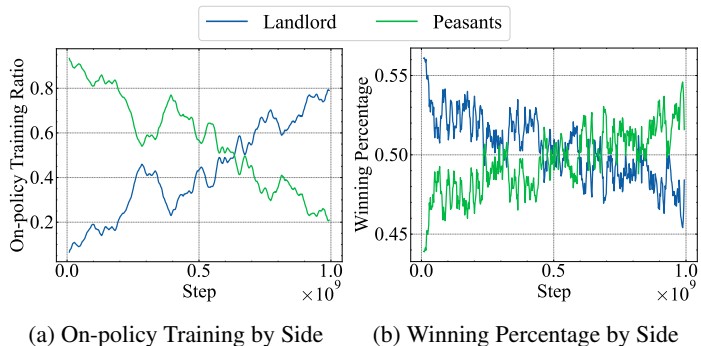

|     |     |
| --- | --- |
| (a) On-policy Training by Side | (b) Winning Percentage by Side |

Figure 14: More training is allocated to the Peasants at the beginning, then gradually shifts to the Landlord as the Peasants become stronger through cooperation.

### G.5 FPDou with adaptive threshold

In the main text, we adopt a fixed threshold $\tau = 0.5$ to guide the $\epsilon_t$-best response learning for both the Landlord and Peasants. This design relies on the assumption that DouDizhu is a balanced game. However, observations from our ablation experiments (see Fig. 3 (b,c) and Fig. 14) reveal a key phenomenon: strength shifts between the two sides over the course of training. Specifically, more on-policy training is initially allocated to the Peasants, then gradually shifts to the Landlord as the Peasants become stronger through cooperation.

This shift reflects an underlying imbalance of DouDizhu. In the early training stages, the Landlord maintains a dominant advantage over the Peasants. As training progresses, the Peasants master effective cooperative strategies and eventually outperform the Landlord. Such dynamic imbalance indicates that the difficulty of achieving a 0.5 win rate differs between the Landlord and Peasants, implying the game may not be balanced and providing a rationale for introducing an adaptive threshold mechanism.

#### G.5.1 Adaptive Threshold

We design an adaptive threshold to account for side-specific training difficulty, using on-policy training time as a proxy for this difficulty. The core intuition is as follows: if one side requires significantly more on-policy training time to approach the target win rate, it implies higher difficulty in reaching the predefined threshold for that side. While on-policy training time is not a perfect metric—factors like algorithmic differences (e.g., the Monte Carlo method employed for both roles may not be equally optimal for each) can affect training efficiency—it remains a practical and reasonable indicator of relative difficulty between the two sides.

Specifically, the adaptive threshold update rule is implemented as follows: (1) Set the initial threshold for both the Landlord and Peasants to 0.5. (2) For each one-hour training window, record the on-policy training time for the Landlord ($T_{\text{Landlord}}$) and Peasants ($T_{\text{Peasants}}$). (3) If one side's on-policy training time is twice or more that of the other side, reduce its threshold by 0.01. For example: if $T_{\text{Landlord}} \geq 40$ minutes (i.e. $T_{\text{Peasants}} \leq 20$ minutes) in a training window, the Landlord's threshold is adjusted to $\tau_{\text{Landlord}} = 0.5 - 0.01 = 0.49$, while the Peasants' threshold is adjusted to $\tau_{\text{Peasants}} = 0.5 + 0.01 = 0.51$. In this case, the Landlord only needs to achieve a 0.49 win rate to qualify as an $\epsilon_t$-best response, whereas the Peasants must meet the higher 0.51 threshold.

#### G.5.2 Results

We first present the threshold values during training in Fig. 15. For the Landlord, we observe that the threshold increases to 0.53 at the early stage of training and then drops to 0.46 afterward. This indicates that initially, the Landlord easily defeats the Peasants, leading to a higher threshold. As training progresses, the Peasants become stronger than the Landlord, causing the Landlord's threshold to decrease. This observation aligns with the results from our main experiment with a fixed threshold, where the Landlord is stronger at the beginning while the Peasants gain strength later, as shown in Figs. 3b, 3c and 14.

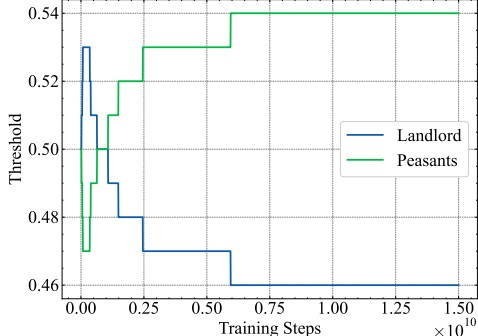

Figure 15: The win rate threshold during training. For the Landlord, the threshold increases to 0.53 briefly at the beginning, then decreases to 0.46.

Next, from the learning curves in Figs. 16 and 17, we find patterns similar to those in Figs. 9 and 10. Both fixed and adaptive thresholds yield stable performance, with no sharp drops during training.

Additionally, minor differences emerge due to the threshold mechanism: the adaptive threshold results in a slightly lower Landlord win rate but a slightly higher Peasant win rate. The comparison against SL provides a clear example: FPDou (fixed threshold) achieves an approximate 0.6 win rate as the Landlord, while FPDou with adaptive threshold reaches around 0.59. In contrast, FPDou (fixed threshold) has an approximate 0.77 win rate as the Peasants, compared to 0.78 for the adaptive-threshold variant. These differences are minimal—especially when competing against strong methods like DouZero and PerfectDou, where negligible performance gaps exist between the two variants. Despite the slight differences, this still demonstrates that threshold settings impact performance, which could be explored in future work.

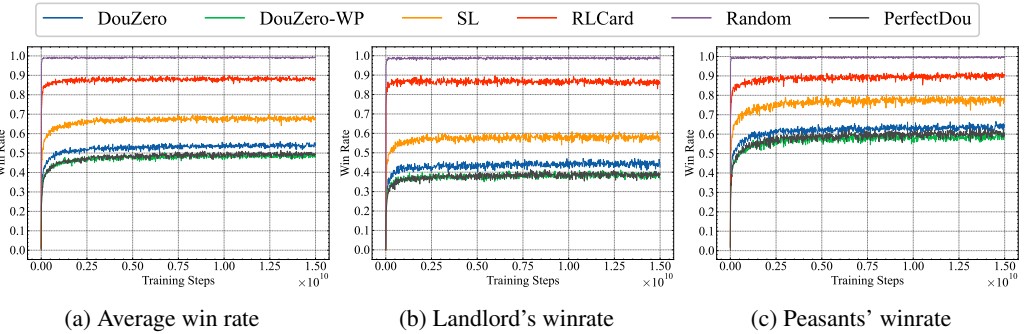

(a) Average win rate      (b) Landlord's winrate      (c) Peasants' winrate

Figure 16: Win rate (WP) learning curves of FPDou (with adaptive threshold) against baseline methods.

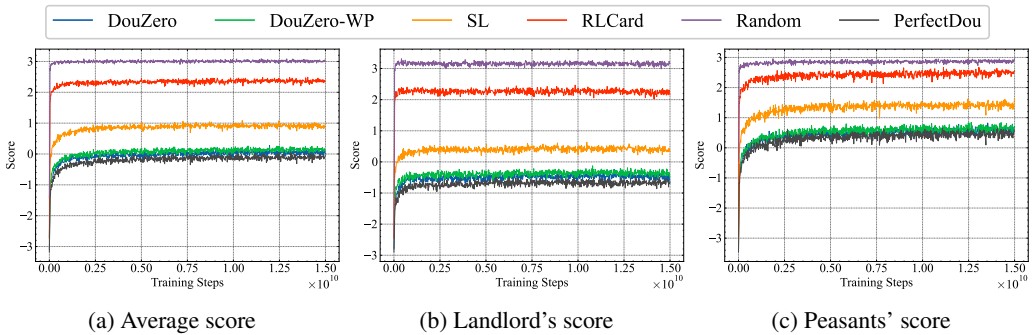

(a) Average score      (b) Landlord's score      (c) Peasants' score

Figure 17: Score (ADP) learning curves of FPDou (with adaptive threshold) against baseline methods.

