# OpenReview forum: "FPDou: Mastering DouDizhu with Fictitious Play"
_ICLR.cc/2026/Conference — Submitted to ICLR 2026_

### Official Review · Reviewer_pnAj · 2025-10-31

**Soundness:** 2
**Presentation:** 1
**Contribution:** 3
**Rating:** 2
**Confidence:** 3

**Summary:**

The authors present FPDou, which achieves SOTA performance in DouDizhu using an extension of generalized weakened fictitious self play. In training, DouDizhu is treated as a two-player zero-sum game where the two peasants are treated as a team sharing perfect information. To extend the game to 3 players at deployment with full imperfect information, partial-observation feature extractors are trained for the peasants to have similar features as the perfect information encoders. These imperfect information features are then used in deployment. FPDou outperforms agents from prior work in skill.

**Strengths:**

The paper presents an impressive and difficult to achieve empirical result, attaining SOTA performance in DouDizhu.

**Weaknesses:**

While the main empirical result is very impressive, there are several issues in my opinion with the communication of the method. Revising the paper with clearer language would help address many of my concerns.

1. The presentation of the proposed approach is unclear. In line 203, it's claimed that a mixture of deep RL and supervised learning is used to learn a $\epsilon$-best response as well as the average policy, yet this supervised learning component of the optimization is never mentioned again. Where in Algorithm 1 or the loss functions in section 4 does the method ensure that an average policy is learned? How is the average-policy supervised learning facilitated, and what loss is used? The theoretical explanation in section 3.2 follows, but it isn't clear how the process in the last paragraph of section 3.2 is actually implemented, which seems to be a key detail.

2. The proposed approach to make partially observed features similar to fully observed features is a heuristic. The paper relies on an L2 feature regularize to “recover” the imperfect-info policy. This is a concise solution but somewhat ad-hoc. Unless there is a properly specific to DouDizhu that allows it, there’s no guarantee that an NE with imperfect-information is similar to an NE with teammate-shared perfect information. It would improve the paper if the authors could please clarify any convergence implications or limitations of using perfect-info training and imperfect-info execution.


I believe a missing limitation is that because the peasant policies are trained to only work with each other, they may not effectively cooperate well in ad-hoc peasant teams with other players/policies. (This limitation itself is not a weakness)

Also see Questions, which concern unclear aspects of the method.

**Questions:**

a) Unless I am confused, the distinction between on-policy and off-policy seems to misuse vocabulary. According to Algorithm 1, the off-policy Q-learning method is used in all stages, always drawing data from an (off-policy) replay buffer $\mathcal{D}$. To my understanding, at no point is "on-policy" RL ever actually used. Would a more appropriate terminology be something like off-policy learning with staggered opponent freezing?

b) The replay buffer size of 100,000 seems very small for keeping historical data from all past best responses. How are you ejecting data from this replay buffer? Are you using reservoir sampling like in NFSP [1]? How does this replay buffer ensure that you can produce an average policy over a long training history across many days?

c) Section E.4: What is "Fraction of off-policy data in each batch: 0.5"? I don't see explicit mention of this elsewhere in the paper. Are there two sources of data from which batches are constructed rather than the single replay buffer used in line 14 of Algorithm 1?

[1] Heinrich, J., & Silver, D. (2016). Deep Reinforcement Learning from Self-Play in Imperfect-Information Games.

**Details Of Ethics Concerns:**

No ethics concerns.

---

> ### Author Response · Authors · 2025-11-20
>
> We would like to sincerely express our gratitude to the reviewer for their time and effort in reviewing our paper. We address each concern you raised and are happy to answer any further questions.
>
> > **Weakness1.** The presentation of the proposed approach is unclear. In line 203, it's claimed that a mixture of deep RL and supervised learning is used to learn a $\epsilon$-best response as well as the average policy, yet this supervised learning component of the optimization is never mentioned again. Where in Algorithm 1 or the loss functions in section 4 does the method ensure that an average policy is learned? How is the average-policy supervised learning facilitated, and what loss is used? The theoretical explanation in section 3.2 follows, but it isn't clear how the process in the last paragraph of section 3.2 is actually implemented, which seems to be a key detail.
>
> The entire process is based on training a Q-network, which maps state-action pairs to final returns. Specifically, when using data from current interactions, updating Q-network, and collecting new data, we form a policy iteration loop. This is what we refer to as deep RL. When sampling data from the replay buffer (which store past experience), the network performs what we refer to as supervised learning: it fits the average of past $\epsilon$-best responses. Thus, the two learning processes are unified by the Q-network and are updated using the same loss function (Eq.8).
>
> The two learning processes are controlled by the hyperparameter "Fraction of off-policy data in each batch" (Table 6), which we set to 0.5. This means each training batch comprises 50% recent on-policy interaction data for the "best-response" RL component, and 50% historical data for the "average-strategy" SL component. We added more details in Section 3.3 and 4.2 to clarify this aspect.
>
> > **Weakness2.** The proposed approach to make partially observed features similar to fully observed features is a heuristic. The paper relies on an L2 feature regularize to “recover” the imperfect-info policy. This is a concise solution but somewhat ad-hoc. Unless there is a properly specific to DouDizhu that allows it, there’s no guarantee that an NE with imperfect-information is similar to an NE with teammate-shared perfect information. It would improve the paper if the authors could please clarify any convergence implications or limitations of using perfect-info training and imperfect-info execution.
>
> The ultimate objective for peasants in DouDizhu is identical and highly aligned: to cooperatively defeat the Landlord. The primary challenge under imperfect information lies in coordination. The perfect-information policy, therefore, serves as a powerful standard for ideal coordination. Our approach is based on the intuition that a strong imperfect-information policy should strive to approximate the decisions of this ideal cooperative policy. Our L2 regularization is the direct implementation of this intuition: it leverages the latent representation of the perfect-information policy as a teacher signal, guiding the latent representation of the imperfect-information policy toward decisions most conducive to cooperative optimality. Experimental results demonstrate the practical effectiveness of this heuristic, and its utility is further validated by the success of prior work such as OADMCDou [1].
>
> > **Weakness3.** I believe a missing limitation is that because the peasant policies are trained to only work with each other, they may not effectively cooperate well in ad-hoc peasant teams with other players/policies. (This limitation itself is not a weakness)
>
> Two peasants forming a fixed team is a formal setup in official DouDizhu competitions, thus we focus on training two well-coordinated teammates. The proposed zero-shot coordination, which exists in casual human play is out of the scope of our current work. We have discussed this interesting direction in the Limitations section of our manuscript. We thank the reviewer for helping us provide a more complete and nuanced understanding of our method's potential boundaries.

---

> > ### Author Response · Authors · 2025-11-20
> >
> > > **Question.** a) Unless I am confused, the distinction between on-policy and off-policy seems to misuse vocabulary. According to Algorithm 1, the off-policy Q-learning method is used in all stages, always drawing data from an (off-policy) replay buffer $\mathcal{D}$. To my understanding, at no point is "on-policy" RL ever actually used. Would a more appropriate terminology be something like off-policy learning with staggered opponent freezing?
> >
> > For the agent designated as the "on-policy" learner, we use the trajectories it generates directly through environmental interactions, and conduct an update. This iterative loop is the "on-policy" aspect of our framework. For the agent designated as the "off-policy" learner, which is a copy of the fixed opponent, its learning relies on data not generated by itself. This is the "off-policy" aspect. Therefore, our terminology distinguishes between the learning signal from immediate, self-generated data (on-policy) and that from data generated by other policies (off-policy). Both processes share the same Monte Carlo Estimation implementation, with the sole difference being whether we synchronize the learner’s parameters with the data-generating actors following each update. We have revised Algorithm 1 to better clarify the learning processes.
> >
> > > **Question.** b) The replay buffer size of 100,000 seems very small for keeping historical data from all past best responses. How are you ejecting data from this replay buffer? Are you using reservoir sampling like in NFSP? How does this replay buffer ensure that you can produce an average policy over a long training history across many days?
> >
> > We would like to clarify that we store entire game trajectories as single units in our buffer, with a capacity of 100,000 games. In comparison, the large-scale AlphaGo Zero system [2] was trained using the most recent 500,000 games, thus our buffer may not be very small. Currently, we use a simple First-In-First-Out (FIFO) buffer. We agree that adopting a method like reservoir sampling would better align with the theoretical principle of averaging over the entire play history, we have included this direction in the Limitations section as future work.
> >
> > > **Question.** c) Section E.4: What is "Fraction of off-policy data in each batch: 0.5"? I don't see explicit mention of this elsewhere in the paper. Are there two sources of data from which batches are constructed rather than the single replay buffer used in line 14 of Algorithm 1?
> >
> > The hyperparameter controls the composition of each training batch. A value of 0.5 means each batch comprises two data sources: 50% is sampled from the replay buffer, which contains a rich history of past experiences. The other 50% consists of data generated from the most recent environmental interactions. As clarified in our response to Weakness 1, this serves as the practical mechanism underpinning our unified learning framework. The data from recent interactions is used to learn the new $\epsilon$-best response (the RL component), while the data from the historical replay buffer is used to learn the average of past $\epsilon$-best responses (the SL component).
> >
> > [1] Luo, Qian, et al. "Enhanced DouDiZhu card game strategy using oracle guiding and adaptive deep Monte Carlo method." Proceedings of the Thirty-Third International Joint Conference on Artificial Intelligence. 2024.
> >
> > [2] Silver, David, et al. "Mastering the game of go without human knowledge." nature 550.7676 (2017): 354-359.

---

> ### Author Response · Authors · 2025-11-27
> **Gentle Reminder**
>
> Dear Reviewer pnAj,
>
> Thank you again for your constructive comments and the time you dedicated to reviewing our paper.
> As the discussion period is approaching its end, we would appreciate it if you could let us know whether our responses have sufficiently addressed your concerns or if there are any remaining issues we can clarify.
>
> We look forward to your feedback.
>
> Best regards,
>
> The Authors

---

### Official Review · Reviewer_EC6D · 2025-11-01

**Soundness:** 2
**Presentation:** 2
**Contribution:** 2
**Rating:** 2
**Confidence:** 3

**Summary:**

This paper introduces an algorithm for learning to play DouDizhu. It is an amalgamation of ideas for building a new SOTA.

**Strengths:**

The paper does an extensive evaluation of the proposed algorithm. There many tasteful practical choices made, from observing that the reward is discrete and using distributional RL to balancing training with win percentage.

**Weaknesses:**

To start with text itself, there are lots of choice of words that do not make reading the paper easier.
- 038 surely advancements have not been introduced to the game itself
- 050 "this reduces DouDizhu to two-plahyer zero-sum" I think that a noun is missing
- The appendix conflates the literature on extensive form games and game theory. In particular (@ 899) game theory does not assume perfect recall.
- The paper refers to sequence form strategies as strategies.
- At this level, introducing Kuhn's theorem is odd and it is not clear why realization-equivalent (I think it should be equivalence) is introduced.
- The derivation of the GWFP (both in appendix and the main text) takes a lot of space just to make the point that sampling from the average sequence form policy is equivalent to first sampling a sequence form and then sampling from that sequence form policy.
- 1278 the sum diverges and does not converge to zero.
- The paper if fixated on being theoretically correct and grounded in GWFP but conveniently ignores that the perfect training imperfect execution (PTIE) paradigm is clearly not safe. The paper that introduced PTIE claims that it is an extension of centralized training, distributed execution but this is clearly not the case, the value net (the central element of training) is not used in execution but the Q functions training with PTIE are distilled into the average policy.
- Going to the contributions of the paper, the only substantial contribution seems to be the off-policy, on-policy flags introduced in the loop
   -  GWFP has been used for this game, so has PTIE, and learning sequence form averages from replay bufer (cf PSRO)
- Twice, the paper claims that policy churns helps the exploration but policy churn is omnipresent everywhere so what makes DouDizhu special?

**Questions:**

Figure 12 a is the sampling strategy used to retrain a network from scratch? If so how long did the training take, how does the choice of temperature affects the WP?
Can you explain ADP again?
Table 2: Can the error be estimated with bootstrap?

---

> ### Author Response · Authors · 2025-11-20
>
> We would like to sincerely express our gratitude to the reviewer for their time and effort in reviewing our paper. We address each concern you raised and are happy to answer any further questions.
>
> > **Weakness1.** 038 surely advancements have not been introduced to the game itself.
> >
> > **Weakness2.** 050 "this reduces DouDizhu to two-plahyer zero-sum" I think that a noun is missing
> >
> > **Weakness3.** The appendix conflates the literature on extensive form games and game theory. In particular (@ 899) game theory does not assume perfect recall.
> >
> > **Weakness4.** The paper refers to sequence form strategies as strategies.
> >
> > **Weakness5.** At this level, introducing Kuhn's theorem is odd and it is not clear why realization-equivalent (I think it should be equivalence) is introduced.
>
> We sincerely thank the reviewer for their meticulous reading and detailed feedback, which has helped us significantly improve the precision and clarity of our manuscript. All raised points have been addressed in the revised version of the paper.
>
> Regarding the precision of our phrasing (Weaknesses 1 & 2): We have rectified the inaccuracies ("introduced to mastering the game","a two-player zero-sum game") in the introduction.
>
> Regarding the theoretical background (Weaknesses 3, 4, & 5): We appreciate the reviewer pointing out the lack of precision in our theoretical discussion. We have performed a thorough revision of section Appendix D.1 to address all concerns raised: 1) We now clearly distinguish between the broad field of game theory and the specific context of extensive-form games, clarifying that perfect recall is a key assumption for the latter. 2) We have carefully reviewed our use of the term "strategy" to ensure precision and clarified that our work uses behavioral strategies, not sequence-form strategies. 3) Most importantly, we have entirely rewritten the relevant section to explicitly state the rationale for introducing Kuhn's Theorem and realization equivalence. The revised text now clearly presents these concepts as the essential theoretical bridge that connects the mixed-strategy framework of Fictitious Play with our practical, behavioral-strategy-based deep RL implementation.
>
> > **Weakness6.** The derivation of the GWFP (both in appendix and the main text) takes a lot of space just to make the point that sampling from the average sequence form policy is equivalent to first sampling a sequence form and then sampling from that sequence form policy.
>
> The primary purpose of the derivation in Section 3.2 is to provide the theoretical justification for our key algorithmic contribution: simplifying the two-network Fictitious Play implementation into a highly efficient one-network deep RL algorithm. The primitive form reveals how the two distinct processes (learning a new best response and averaging past responses) can be unified. This insight is precisely what enables the use of a single network to accomplish both processes simultaneously via a replay buffer. Following the reviewer's suggestion, we have significantly condensed Section 3.2 in our revised manuscript. The new version de-emphasizes the derivation itself and focuses on how this theoretical insight inspired our simplification from a two-network to a one-network design. The full derivation is retained in the appendix for completeness.
>
> > **Weakness7.** 1278 the sum diverges and does not converge to zero.
>
> We thank the reviewer again for their rigor, which has greatly enhanced the technical soundness of our paper. In the revised manuscript, we have replaced the flawed argument in Appendix F with a rigorous proof that formally establishes the satisfaction of GWFP’s perturbation condition under standard stochastic approximation assumptions.
>
> > **Weakness8.** The paper if fixated on being theoretically correct and grounded in GWFP but conveniently ignores that the perfect training imperfect execution (PTIE) paradigm is clearly not safe. The paper that introduced PTIE claims that it is an extension of centralized training, distributed execution but this is clearly not the case, the value net (the central element of training) is not used in execution but the Q functions training with PTIE are distilled into the average policy.
>
> The use of PTIE in our work is motivated by its empirical success in prior SOTA agent PerfectDou [1].To address your concerns with maximum accuracy, we would greatly appreciate some clarification. Regarding the claim that PTIE is "not safe", could the reviewer kindly elaborate on the specific risks they are referring to? Additionally, our paper does not discuss Centralized Training, Distributed Execution. We would also be grateful if the reviewer could provide more context on this point. We look forward to providing a more detailed response later.

---

> > ### Author Response · Authors · 2025-11-20
> >
> > > **Weakness9.** Going to the contributions of the paper, the only substantial contribution seems to be the off-policy, on-policy flags introduced in the loop. GWFP has been used for this game, so has PTIE, and learning sequence form averages from replay bufer (cf PSRO)
> >
> > Our contribution is not merely about "off-policy, on-policy flags", but the development of a novel method that successfully bridges the gap between the rigorous theory of GWFP and the practical demands of large-scale deep reinforcement learning in a notoriously challenging task DouDizhu.
> >
> > Regarding GWFP: While prior works on DouDizhu are inspired by self-play, to our knowledge, our work is the first to propose a practical DRL method that is both theoretically sound and empirically efficacious. Previous methods typically update all players simultaneously, which violates the stationarity assumption (and other key conditions) required for guaranteed best-response learning. Our framework is explicitly designed to preserve this theoretical guarantee.
> >
> > Regarding PTIE: We acknowledge the prior use of PTIE. Our contribution here lies in a novel and more efficient approach to implementing PTIE in a value-based setting. Instead of requiring two distinct networks (e.g., an Actor and a Critic), our latent representation regularization achieves this with a single, unified network.
> >
> > Regarding learning averages from a replay buffer (cf. PSRO): While the core idea of learning from past policies is central to many self-play algorithms (including PSRO), our key innovation lies in the efficient implementation of this principle. By leveraging a standard replay buffer, our method elegantly consolidates the processes of best-response learning and historical averaging into the update of a single network, drastically simplifying the algorithm relative to prior methods.
> >
> > Ultimately, the success of this novel theory-practice integration is embodied in our core contribution: a new SOTA agent for DouDizhu. We are confident in the strength of our method and have provided both an open-source implementation and an interactive demo. We respectfully invite the reviewer to test our agent to experience its performance firsthand.
> >
> > > **Weakness10.** Twice, the paper claims that policy churns helps the exploration but policy churn is omnipresent everywhere so what makes DouDizhu special?
> >
> > First, as shown in the original policy churn paper ([3], Figure 2), policy churn alone, without any explicit exploration, can achieve performance comparable to $\epsilon$-greedy strategies in value-based RL. This provides compelling evidence that natural policy drift during updates is itself a powerful implicit exploration mechanism.
> >
> > Second, the extensive initial state diversity of DouDizhu amplifies the effectiveness of this implicit exploration. Unlike games with fixed starting positions like Go or Atari, every episode of DouDizhu begins with a novel, randomly dealt hand. This provides immense built-in exploration from the first step. This is consistent with findings from AlphaGo, where exploration (via Dirichlet noise) was focused primarily on the game’s early stages [4,5]. This suggests that as a game progresses and the state space narrows, the need for broad explicit exploration diminishes.
> >
> > In summary, the combination of DouDizhu's initial randomness and the demonstrated exploratory capacity of policy churn creates a learning environment where sufficient exploration is provided implicitly. This synergy is what makes our approach of relying on minimal explicit exploration both viable and effective.
> >
> > > **Question1.** Figure 12 a is the sampling strategy used to retrain a network from scratch? If so how long did the training take, how does the choice of temperature affects the WP?
> >
> > Yes, each exploration strategy was used to train a separate agent entirely from scratch. Each of these training runs took approximately two days on our hardware.
> >
> > The greedy policy can be viewed as an extreme case of a softmax policy where the temperature approaches zero. Our experimental results show that this greedy policy outperforms the stochastic softmax@n strategy. This result suggests that a lower temperature, leading to less random exploration, is more effective in DouDizhu.
> >
> > > **Question2.** Can you explain ADP again?
> >
> > ADP is a performance metric adopted in previous work[1,2]. The base point is 1. Each bomb doubles the point. For example, No bombs played: winner +1, loser -1; One bomb played: winner +2, loser -2; Two bombs played: winner +4, loser -4.

---

> > > ### Author Response · Authors · 2025-11-20
> > >
> > > > **Question3.** Table 2: Can the error be estimated with bootstrap?
> > >
> > > Following the reviewer's advice, we have performed a bootstrap analysis to compute the standard error (SE) presented in the following table.
> > >
> > > | Rank |   | FPDou   |        | PerfectDou |        | DouZero |        | DouZero-WP |        | SL      |        | RLCard  |        | Random  |        |
> > > |------|-------|---------|--------|------------|--------|---------|--------|------------|--------|---------|--------|---------|--------|---------|--------|
> > > |      |       | WP      | ADP    | WP         | ADP    | WP      | ADP    | WP         | ADP    | WP      | ADP    | WP      | ADP    | WP      | ADP    |
> > > | 1 | FPDou | - | - | **0.520**±0.0050 | **0.100**±0.0262 | **0.562**±0.0050 | **0.197**±0.0262 | **0.510**±0.0050 | **0.333**±0.0268 | **0.684**±0.0048 | **0.996**±0.0245 | **0.894**±0.0032 | **2.522**±0.0231 | **0.993**±0.0008 | **3.107**±0.0178 |
> > > | 2 | PerfectDou | 0.480±0.0049 | -0.100±0.0267 | - | - | **0.543**±0.0050 | **0.141**±0.0260 | 0.489±0.0050 | **0.212**±0.0265 | **0.669**±0.0048 | **1.033**±0.0237 | **0.890**±0.0032 | **2.495**±0.0221 | **0.993**±0.0008 | **3.087**±0.0178 |
> > > | 3 | DouZero | 0.439±0.0049 | -0.197±0.0265 | 0.457±0.0049 | -0.141±0.0266 | - | - | 0.453±0.0049 | **0.119**±0.0263 | **0.611**±0.0050 | **0.774**±0.0249 | **0.857**±0.0035 | **2.377**±0.0233 | **0.987**±0.0011 | **3.043**±0.0183 |
> > > | 4 | DouZero-WP | 0.490±0.0050 | -0.333±0.0250 | **0.511**±0.0050 | -0.212±0.0262 | **0.548**±0.0050 | -0.119±0.0263 | - | - | **0.660**±0.0049 | **0.715**±0.0254 | **0.884**±0.0033 | **2.164**±0.0243 | **0.988**±0.0011 | **2.741**±0.0213 |
> > > | 5 | SL | 0.316±0.0048 | -0.996±0.0247 | 0.331±0.0048 | -1.033±0.0251 | 0.389±0.0048 | -0.774±0.0240 | 0.340±0.0049 | -0.715±0.0248 | - | - | **0.808**±0.0039 | **1.787**±0.0223 | **0.974**±0.0016 | **2.696**±0.0214 |
> > > | 6 | RLCard | 0.106±0.0031 | -2.522±0.0232 | 0.110±0.0031 | -2.495±0.0220 | 0.144±0.0035 | -2.377±0.0233 | 0.116±0.0032 | -2.164±0.0249 | 0.192±0.0041 | -1.787±0.0227 | - | - | **0.942**±0.0024 | **2.504**±0.0227 |
> > > | 7 | Random | 0.007±0.0008 | -3.107±0.0184 | 0.007±0.0008 | -3.087±0.0183 | 0.013±0.0011 | -3.043±0.0186 | 0.012±0.0011 | -2.741±0.0214 | 0.026±0.0016 | -2.696±0.0213 | 0.058±0.0023 | -2.504±0.0222 | - | - |
> > >
> > > [1] Yang, Guan, et al. "Perfectdou: Dominating doudizhu with perfect information distillation." Advances in neural information processing systems 35 (2022): 34954-34965.
> > >
> > > [2] Jiang, Qiqi, et al. "DeltaDou: Expert-level Doudizhu AI through Self-play." IJCAI. 2019.
> > >
> > > [3] Schaul, Tom, et al. "The phenomenon of policy churn." Advances in Neural Information Processing Systems 35 (2022): 2537-2549.
> > >
> > > [4] Silver, David, et al. "Mastering the game of Go with deep neural networks and tree search." nature 529.7587 (2016): 484-489.
> > >
> > > [5] Silver, David, et al. "Mastering the game of go without human knowledge." nature 550.7676 (2017): 354-359.

---

> ### Author Response · Authors · 2025-11-27
> **Gentle Reminder**
>
> Dear Reviewer EC6D,
>
> Thank you again for your constructive comments and the time you dedicated to reviewing our paper.
> As the discussion period is approaching its end, we would appreciate it if you could let us know whether our responses have sufficiently addressed your concerns or if there are any remaining issues we can clarify.
>
> We look forward to your feedback.
>
> Best regards,
>
> The Authors

---

### Official Review · Reviewer_UK6c · 2025-11-02

**Soundness:** 3
**Presentation:** 3
**Contribution:** 2
**Rating:** 6
**Confidence:** 4

**Summary:**

This paper presents FPDou, a new RL framework designed to master the three-player imperfect-information card game DouDizhu. The paper adapts Generalized Weakened Fictitious Play to a deep RL setting, addressing the non-stationarity issue of multi-agent training. The authors convert the three-player game into a two-player zero-sum formulation by treating the two Peasants as a unified agent, and also apply alternative on-policy/off-policy updates and distributional Q-networks. Empirically, FPDou achieves SOTA performace with smaller models.

**Strengths:**

- The idea of the paper is overall well-motivated. The analysis of instability in simultaneous self-play is interesting, and the proposed solution effectively addresses this issue under the given setting, providing valuable insights for the research community.
- The empirical performance is strong — the paper achieves SOTA results using a smaller model, while maintaining a reasonable training cost.
- The writing is clear and easy to follow.

**Weaknesses:**

- The main obstacle to applying GWFP to the DouDizhu problem is its two-player zero-sum game setting, which the paper addresses by merging the peasants and adding a regularization term. However, simply aligning the latent representations with or without perfect information does not eliminate the need for perfect information and lacks a sound rationale. Therefore, FPDou is unlikely to satisfy the PTIE framework, since prior related work used only perfect information during policy evaluation.
- The explanation of the off-policy component of the framework is unclear. If a fixed opponent is required, why is off-policy learning necessary? Which algorithm is used for the updates? How does off-policy learning ensure the stability of the model?
- The paper sets a 0.5 win-rate threshold to ensure the ε-best response, which still introduces a degree of heuristics. There is also a potential risk that, in some iterations, the model may never reach a 0.5 win rate, thereby blocking training. This affects the method's generalizability. Although the authors mention an automated threshold adjustment process in the appendix, there is no noticeable difference in performance, which is somewhat counterintuitive and warrants further explanation.
- I still have concerns regarding the generalizability of the paper, considering that it solely focuses on DouDizhu and many design choices and findings are problem-specific. For instance, are there existing works combining GWFP with RL in other games? How can other multi-agent games be generally transformed into two-player zero-sum settings? Which components of FPDou’s design could provide insights or inspiration for researchers working on different tasks?

**Questions:**

- More information on addressing the training-execution gap should be provided, such as visualizations of the regularization term during training or more effective methods to prevent agent cheating.
- The motivation and methodology for off-policy learning need to be further explained.
- Related work on applying GWFP to other games, as well as more generalizable takeaways, needs to be added.

---

> ### Author Response · Authors · 2025-11-20
>
> We thank the reviewer for the valuable comments and positive assessment of our work. We would like to address the concerns you raised in your review.
>
> > **Weakness1.** The main obstacle to applying GWFP to the DouDizhu problem is its two-player zero-sum game setting, which the paper addresses by merging the peasants and adding a regularization term. However, simply aligning the latent representations with or without perfect information does not eliminate the need for perfect information and lacks a sound rationale. Therefore, FPDou is unlikely to satisfy the PTIE framework, since prior related work used only perfect information during policy evaluation.
>
> The Rationale for Regularizing Latent Representations. Our objective is not to "eliminate" the need for perfect information, but rather to train a network that can infer a near-optimal decision-making basis from imperfect information. To achieve this, we use perfect information as a supervisory signal using an L2 regularization term, to compel $z$ to approximate the ideal latent representation $\bar z$. This regularization is a guided feature-learning process that teaches the network to infer global cooperative intent from purely local information. Similar idea have been explored in OADMCDou [1]. And ablation studies in our Figure 3(a) further validate the effectiveness of this regularization term.
>
> PTIE Framework. The core spirit of the PTIE framework is to leverage perfect information during training to enhance the agent's performance during imperfect execution. PerfectDou [2] implements it by using a oracle critic to guide an imperfect actor. While effective, this approach requires training two separate networks. Our method adopts a more efficient and concise single Q-network architecture. Both methods are valid strategies that leverage perfect information to guide and accelerate learning. Crucially, during execution (testing), FPDou receives absolutely no perfect information. The model relies entirely on the inference capability it developed during training, strictly adhering to the "Imperfect Execution" principle.
>
> > **Weakness2.** The explanation of the off-policy component of the framework is unclear. If a fixed opponent is required, why is off-policy learning necessary? Which algorithm is used for the updates? How does off-policy learning ensure the stability of the model?
>
> The Necessity of Off-Policy Learning with a Fixed Opponent. Our framework maintains two instantiations of each player: an actor player and a learner copy. While the actor player of one side is fixed to ensure a stable training environment for its opponent, its corresponding learner copy is trained off-policy using data generated from gameplay. Without this off-policy component, all data generated would be limited to training only one side (the on-policy player), resulting in the waste of half the accumulated experience. By training the fixed actor’s learner copy off-policy, we leverage every generated trajectory to enhance the performance of both sides concurrently, dramatically accelerating the overall training process (Figure 3(a)).
>
> The Algorithm for Off-Policy Updates. The algorithm used for the off-policy updates is identical to that used for on-policy one: Deep Monte Carlo with a distributional Q-network. The only distinction lies in the data source: for the on-policy player, data is collected from its own interactions with the environment; for the off-policy player, the data is not self-generated.
>
> How Off-Policy Learning Ensures Stability. The stability of our framework is ensured on two levels, with the off-policy component playing a crucial role. First, the primary source of stability is the stationary environment provided for the on-policy learner, which is achieved by fixing the opponent's policy during each training iteration. Second, the off-policy updates themselves are inherently stable. Our method relies on Monte Carlo (MC) returns rather than bootstrapped Temporal-Difference (TD) targets. It avoids a key element of the "deadly triad"—the instability arising from bootstrapping with function approximation [3]. This non-bootstrapping approach to value estimation has also been successfully employed in AlphaGo for its value network [4,5].

---

> > ### Author Response · Authors · 2025-11-20
> >
> > > **Weakness3.** The paper sets a 0.5 win-rate threshold to ensure the ε-best response, which still introduces a degree of heuristics. There is also a potential risk that, in some iterations, the model may never reach a 0.5 win rate, thereby blocking training. This affects the method's generalizability. Although the authors mention an automated threshold adjustment process in the appendix, there is no noticeable difference in performance, which is somewhat counterintuitive and warrants further explanation.
> >
> > Fixed Threshold 0.5. We acknowledge that the 0.5 threshold is a heuristic. However, it is the most natural and principled choice in the absence of prior knowledge about the game's true equilibrium. A 0.5 win rate represents a neutral, unbiased target for an ideal equilibrium, any other fixed value would introduce an arbitrary and less justifiable bias.
> >
> > Potential risk of blocking training. The reviewer correctly notes that the alternation frequency decreases as training progresses. We have also observed this phenomenon (Figure 3 (c)). However, we argue that this is not a training failure or "blockage," but rather an indicator of convergence. This behavior can be formally understood through the lens of exploitability. For example, if the true equilibrium is 0.45 (Landlord) vs. 0.55 (Peasants), our training will naturally stabilize when the Landlord's win rate plateaus around 0.45 and cannot reach the 0.5 target. This outcome implies that the Peasants' policy has become so strong that its exploitability is less than 0.05. Otherwise, the Landlord would learn to exploit the remaining weaknesses and push its win rate to 0.45 + 0.05 = 0.5. Therefore, when the system settles into a state where one player cannot reach the 0.5 threshold, it provides evidence that we have found an $\epsilon$-Nash equilibrium, where $\epsilon$ is bounded by the gap between the stabilized win rate and 0.5. This is not a bug, but a feature that signals convergence. While the ideal objective is the true Nash equilibrium, its corresponding win rate is unknown. The SOTA performance we achieve empirically validates that this resulting $\epsilon$-Nash equilibrium corresponds to a highly proficient and competitive policy.
> >
> > Limited Impact of the Adaptive Threshold. The marginal performance difference from the adaptive threshold is explained by the powerful self-regulating properties inherent to our base framework. First, our framework has an implicit self-adaptive mechanism. With a fixed threshold, the system automatically allocates more on-policy training time to the weaker player (the one failing to meet the threshold). Second, and critically, the off-policy player is never idle. While one side is being trained on-policy for an extended period, its opponent is continuously improving via off-policy updates. This ensures a smooth, co-evolutionary process where both sides progress, greatly reducing the system's sensitivity to a perfectly tuned threshold. Together, these two mechanisms make our base model with a fixed threshold robust and efficient, pushing its performance close to the ceiling under our current compute and model constraints, leaving little room for an adaptive threshold to provide significant additional benefit.

---

> > > ### Author Response · Authors · 2025-11-20
> > >
> > > > **Weakness4.** I still have concerns regarding the generalizability of the paper, considering that it solely focuses on DouDizhu and many design choices and findings are problem-specific. For instance, are there existing works combining GWFP with RL in other games? How can other multi-agent games be generally transformed into two-player zero-sum settings? Which components of FPDou’s design could provide insights or inspiration for researchers working on different tasks?
> > >
> > > We thank the reviewer for this question about the generalizability of our work. While our goal is to obtain a SOTA doudizhu AI, we believe the underlying principles and several key components of our design are broadly applicable and offer valuable insights for the multi-agent RL community.
> > >
> > > The Combination of GWFP and RL in Other Games. The paradigm of combining Fictitious Play with deep RL is not unique to our work, such as NFSP on Leduc Hold’em [6] and PFSP on StarCraft II [7].
> > >
> > > Transform into two-player zero-sum settings. The transformation of a multi-agent game into a two-player setting is applicable to a wide and important class of games characterized by fixed alliances. The key condition is that a subset of players shares a common objective and a perfectly correlated reward function. Any N-vs-N team game, such as Bridge (2v2), League of Legends (5v5), or Counter-Strike (5v5), can be naturally abstracted into a two-player game between Team A and Team B. We acknowledge this transformation is not universally applicable to games with dynamic alliances or free-for-all dynamics (e.g., Diplomacy). However, it provides a powerful and principled simplification for a vast range of popular multi-agent settings.
> > >
> > > Insights from FPDou's Design. Beyond the specific application, FPDou introduces several generalizable design patterns for tackling core challenges in multi-agent RL.
> > > - The Alternating On/Off-Policy Training Scheme. Our framework for ensuring a stationary environment (via a fixed on-policy opponent) while maximizing sample efficiency (via an off-policy copy) is a highly transferable methodology. It offers a solution to the stability-efficiency trade-off in any self-play setting, potentially doubling sample efficiency without compromising convergence guarantees.
> > > - PTIE via Latent Representation Regularization. Our method of using a perfect-information view to regularize the latent features of a network is a novel, general technique for implementing the PTIE framework within a value-based setting. It provides an alternative to the standard two-network Actor-Critic approach.
> > > - A Paradigm for Efficiently Implementing Policy Averaging. Algorithms like Fictitious Play often require averaging over a history of policies, a process that is computationally prohibitive with large neural networks. FPDou introduces an efficient paradigm to solve this by merging two distinct theoretical steps—learning a new best response and averaging over past policies—into the update of a single network. This is accomplished by learning from a replay buffer, where the data from historical policies serves as an efficient proxy for the policies themselves. This principle of unifying the best-response calculation and historical averaging offers a practical implementation for other complex games in large-scale deep RL.
> > >
> > > > **Question1.** More information on addressing the training-execution gap should be provided, such as visualizations of the regularization term during training or more effective methods to prevent agent cheating.
> > >
> > > We added a visualization of the L2 regularization loss term ($\|z - \bar z\|^2$) over the course of training to the appendix of the final manuscript in Figure 7. The regularization loss is high at the beginning of training, as the randomly initialized network produces divergent representations for perfect ($\bar z$) and imperfect ($z$) information states. As training progresses, the loss decreases steadily and converges to a low, stable value. This trend provides strong empirical validation for our method, demonstrating that the regularization term is effective at actively forcing the network to align its latent representations.
> > >
> > > Agent cheating. Our agent has no access to any perfect information when taking actions during data collection and evaluation, the perfect information is only used as a regularization term during network training. Therefore, our agent does not cheat.
> > >
> > > > **Question2.** The motivation and methodology for off-policy learning need to be further explained.
> > >
> > > Thank you. We hope our detailed response to **Weakness2.** has fully addressed this concern. The key points are that the off-policy component is crucial for sample efficiency, while stability is ensured by our use of non-bootstrapping Monte Carlo returns.

---

> > > > ### Author Response · Authors · 2025-11-20
> > > >
> > > > > **Question3.** Related work on applying GWFP to other games, as well as more generalizable takeaways, needs to be added.
> > > >
> > > > We thank the reviewer for this suggestion. As detailed in our response to **Weakness4.**, our work has strong connections to the broader literature and offers several generalizable takeaways. Related Work: We discussed how the principles of Fictitious Play and RL are applied in works ranging from NFSP in Poker to large-scale agents like AlphaStar in StarCraft II. Three Generalizable Takeaways: (1) The Alternating On/Off-Policy Training Scheme for balancing stability and sample efficiency. (2) PTIE via Latent Representation Regularization as a novel implementation for value-based methods. (3) A Paradigm for Efficiently Implementing Game-Theoretic Averaging using a single network and a replay buffer. If the reviewer finds this discussion satisfactory, we will incorporate these points into the Related Work and Conclusion sections of the final manuscript to make the broader contributions of our work more explicit.
> > > >
> > > > [1] Luo, Qian, et al. "Enhanced DouDiZhu card game strategy using oracle guiding and adaptive deep Monte Carlo method." Proceedings of the Thirty-Third International Joint Conference on Artificial Intelligence. 2024.
> > > >
> > > > [2] Yang, Guan, et al. "Perfectdou: Dominating doudizhu with perfect information distillation." Advances in neural information processing systems 35 (2022): 34954-34965.
> > > >
> > > > [3] Sutton, Richard S., and Andrew G. Barto. Reinforcement learning: An introduction. Vol. 1. No. 1. Cambridge: MIT press, 1998.
> > > >
> > > > [4] Silver, David, et al. "Mastering the game of Go with deep neural networks and tree search." nature 529.7587 (2016): 484-489.
> > > >
> > > > [5] Silver, David, et al. "Mastering the game of go without human knowledge." nature 550.7676 (2017): 354-359.
> > > >
> > > > [6] Heinrich, Johannes, and David Silver. "Deep reinforcement learning from self-play in imperfect-information games." arXiv preprint arXiv:1603.01121 (2016).
> > > >
> > > > [7] Vinyals, Oriol, Igor Babuschkin, Wojciech M. Czarnecki, Michaël Mathieu, Andrew Dudzik, Junyoung Chung, David H. Choi et al. "Grandmaster level in StarCraft II using multi-agent reinforcement learning." nature 575, no. 7782 (2019): 350-354.

---

> ### Author Response · Authors · 2025-11-27
> **Gentle Reminder**
>
> Dear Reviewer UK6c,
>
> Thank you again for your constructive comments and the time you dedicated to reviewing our paper.
> As the discussion period is approaching its end, we would appreciate it if you could let us know whether our responses have sufficiently addressed your concerns or if there are any remaining issues we can clarify.
>
> We look forward to your feedback.
>
> Best regards,
>
> The Authors

---

### Author Response · Authors · 2025-12-01
**Global Response**

We thank all reviewers for their thoughtful comments and constructive criticism. We are encouraged by the reviewers' recognition of our algorithm's **"tasteful practical choices"** (EC6D) and **SOTA performance** (UK6c,pnAj).

We have uploaded a revised manuscript incorporating these suggestions. Below, we summarize the major revisions and clarifications made during the rebuttal phase.

**1. Theoretical Rigor and Precision (Addressing EC6D)**

   - **Rewritten Appendix D.1**: We now rigorously distinguish between general Game Theory and Extensive-Form Games, explicitly clarifying the assumption of perfect recall. We also clarified the role of Kuhn’s Theorem and realization equivalence as the bridge connecting Fictitious Play (mixed strategies) to our Deep RL implementation (behavioral strategies).

   - **Corrected Proofs**: We replaced the flawed argument in Appendix F with a rigorous proof demonstrating that GWFP’s perturbation condition is satisfied under standard stochastic approximation assumptions.

   - **Condensed Section 3.2**: We simplified the derivation of GWFP to focus on its role in justifying our single-network architecture, moving the full derivation to the appendix.

**2. Methodological Clarifications (Addressing pnAj & UK6c)**

   - **Unified Learning Framework**: We clarified (Sections 3.3 & 4.2) that our single Q-network simultaneously learns the $\epsilon$-best response (via recent interaction data) and approximates the average strategy (via historical replay buffer data). This is controlled by a 50/50 mixing batch, efficiently implementing GWFP without separate networks.

   - **On/Off-Policy Distinction**: We refined Algorithm 1 and the text to define On-policy as learning from immediate self-generated interactions (best response), and Off-policy as learning from the fixed opponent's generated data (improving efficiency). Off-policy refers to the learner copy of the fixed player updating its parameters using the trajectories that are not self-generated.

   - **PTIE Rationale**: We elaborated on the L2 regularization term. To address Reviewer UK6c’s query, we added Figure 7 in the appendix, visualizing the convergence of the regularization loss, empirically proving that the network effectively learns to infer global cooperative intent from local observations.

**3. Empirical Robustness (Addressing EC6D)**

   - **Bootstrap Analysis**: We performed a bootstrap analysis to compute standard errors for our main results (Table 2), confirming that FPDou's performance advantage over SOTA methods (PerfectDou, DouZero) is statistically significant.

   - **Exploration**: We clarified that the policy churn combined with DouDizhu's high initial state diversity provides sufficient implicit exploration, rendering heavy explicit exploration unnecessary.

**4. Generalizability (Addressing UK6c)**

We have expanded the discussion on the broader impact of FPDou. Our framework offers generalizable insights for Multi-Agent RL beyond DouDizhu, specifically:

   - **The Alternating On/Off-Policy Scheme** for balancing stability and sample efficiency.

   - **Latent Representation Regularization** as a concise, single-network implementation of the PTIE framework.

   - **A paradigm for efficient Game-Theoretic Averaging** using a replay buffer in value-based RL.

We believe these revisions address the reviewers' concerns and strengthen the paper significantly. We are confident that FPDou represents a solid step forward in solving complex imperfect-information games.

Best regards,

The Authors

---

### Meta-Review · Area_Chair_YG1Z · 2026-01-06

**Summary:**

Current reinforcement learning methods for the card game DouDizhu struggle because they update all three players simultaneously, which creates instability and prevents the AI from discovering optimal strategies. The paper introduces FPDou, which treats the game as a two-player competition during training and employs a special updating approach where players take turns improving. In my opinion, the reviewers did a very thorough job and have presented salient arguments about the suitability of this paper for ICLR in its current form. As pointed out by two reviewers, the presentation is weak, uses some terms is potentially confusing ways,  and leaves too many details open such as how to use the reply buffer exactly. One reviewer is saying that the contributions on the RL side are not strong enough, and another one is pointing out that the parts of the approaches presented are of heuristic nature only. While this is fine, it is unclear whether the heuristic also works fine in other games.

**Reviewer Concerns:**

The rebuttal addressed many of the raised issues but did not completely resolve them. The revised version requires a second round of review to properly assess the improved writing quality and, more importantly, to evaluate the degree of novelty in the contributions.

**Reviewer Scores:**

I do not think the reviewers would have changed their scores based on the rebuttal, as the fundamental concerns about novelty and generalizability remain unresolved in the current form.

---

### Decision · Program_Chairs · 2026-01-26

Reject